# Multi-step nucleation pathway of C-S-H during cement hydration from atomistic simulations

Xabier M. Aretxabaleta [1] ✉, Jon López-Zorrilla [1], Iñigo Etxebarria [1,2] & Hegoi Manzano [1] ✉

The Calcium Silicate Hydrate (C-S-H) nucleation is a crucial step during cement hydration and determines to a great extent the rheology, microstructure, and properties of the cement paste. Recent evidence indicates that the C-S-H nucleation involves at least two steps, yet the underlying atomic scale mechanism, the nature of the primary particles and their stability, or how they merge/aggregate to form larger structures is unknown. In this work, we use atomistic simulation methods, specifically DFT, evolutionary algorithms (EA), and Molecular Dynamics (MD), to investigate the structure and formation of C-S-H primary particles (PPs) from the ions in solution, and then discuss a possible formation pathway for the C-S-H nucleation. Our simulations indicate that even for small sizes the most stable clusters encode C-S-H structural motifs, and we identified a $C_4S_4H_2$ cluster candidate to be the C-S-H basic building block. We suggest a formation path in which small clusters formed by silicate dimers merge into large elongated aggregates. Upon dehydration, the C-S-H basic building blocks can be formed within the aggregates, and eventually crystallize.

Cement is produced worldwide at the astonishing rate of about 150 t $s^{-1}$. It is indispensable for social and technological development, and yet, there are many aspects of this material that we do not understand. One of the most crucial ones is the nucleation and growth of the C-S-H, the main component of the hardened cement paste. C-S-H is the acronym for hydrated calcium silicate, an X-ray amorphous phase of variable composition $(CaO)_x(SiO_2)_y(H_2O)_z$ with a layered structure that resembles the tobermorite family of minerals. On the one hand, the nucleation of the C-S-H plays an important role on the hydration kinetics and hence rheology and work-ability of the cement paste. As C-S-H nucleates (and grows) the saturation index of the pore solution decreases, leading to undersaturated conditions that accelerates calcium silicates' dissolution[1–3]. On the other hand, the nucleation and growth controls the microstructure of the C-S-H gel. The heterogeneous distribution of nuclei[3] determines how the C-S-H will fill the space, governing the pore size distribution and tortuosity. Last but not

least, the nucleation may define the mesoscale structure[4] and composition of the solid C-S-H itself[5].

Recent studies indicate that C-S-H nucleation takes place in at least two steps[6–10]. In contrast to the classical nucleation theory (CNT), multistep nucleation paths involve several (meta)stable states preceding the formation of stable nuclei. The exact number of steps and the nucleation paths are system-dependent, yet there are some common features[11–13] like the existence of thermodynamically stable pre-nucleation clusters (PNCs) or more generally metastable primary particles (PPs). The PNCs are small, formally do not have a phase interface, and should be considered solutes rather than solids[14–16]. Second, the aggregation of PNCs triggers a phase separation into a solvent and dense-liquid-like or amorphous solid regions, the latter crystallizing over time[16,17]. The PNC aggregation has been proposed to be driven by a decrease of entropy upon dehydration[18], a decrease in the solute dynamics[19], a destabilization of clusters due to progressive

[1]Fisika saila, Euskal Herriko Unibertsitatea UPV/EHU, Sarriena Auzoa z/g, 48940 Leioa, Basque Country, Spain. [2]EHU Quantum Center, Euskal Herriko Unibertsitatea, UPV/EHU, Leioa, Spain. ✉e-mail: xabier.mendez@ehu.eus; hegoi.manzano@ehu.eus

chemical reactions or speciation[20,21], or intracluster conversion[22]. Direct experimental observation of PNCs and PPs structure and dynamics is extremely challenging due to the time and size scales. For calcium sulfate, the formation and evolution of PNCs up to the final $(CaSO_4) \cdot xH_2O$ polymorphs has been monitored in situ by X-ray small- and wide-angle scattering[14], and rod-like particles have been reconstructed based on the experiments and MD simulations[23]. For Ca-phosphate and Ca-carbonate, atomistic simulations have also identified the formation of dynamically ordered liquid-like oxyanion polymers (DOLLOPs) formed by ion pairs in solution[18,24]. In the mentioned cases, the PNCs and DOLLOPs do not have the exact structure of the final crystal phases, and it is suggested that a posterior ordering of larger aggregates is what defines the different crystalline polymorphs. However, for barite $(BaSO_4)_4$ compact clusters resembling crystalline motifs have been reported after MD simulations in aqueous environments[25].

In the particular case of C-S-H synthesized from sodium silicate and calcium nitrate solutions[10,26], dynamic light scattering (DLS), small-angle X-ray scattering (SAXS), and transmission electron microscopy (TEM) have shown two well differentiated steps: the formation of "amorphous solid spheroids" of about 50-60 nm that reorganize into C-S-H platelets with time[6-10]. In presence of polycarboxylates, the formation of the spheroids is reported to be similar, yet the crystallization is delayed, attributed to a slower diffusion of water from the spheroid[7]. A higher Ca concentration has shown to accelerate the transformation from the amorphous spheroids to the nanoplatelets[9], which was suggested to occur via dissolution-precipitation of C-S-H. Obtaining further microscopic details remains elusive because of the complex characterization of transient species, which are distinguished by their small size and short lifetimes. Missing information includes (1) to determine which are the dominant species and their characteristics, (2) their stability with respect to the species in solution and with respect to the final solid, (3) how they merge/aggregate to form larger structures, and (4) how the platelet structure emerges from the aggregates. Despite the little understanding of the actual nucleation paths in C-S-H, there is enough evidence from related systems showing that if we can understand it, we might be able to control it. For instance, organic additives like polysaccharides, lignosulfonate, and polyacrylate have been shown to (des)stabilize calcium hydroxide PNCs[27], and $Mg^{2+}$ to retard the crystallization by the stabilization of $CaCO_3$ PNCs[28].

Very few studies have explored ion complexation and PPs using atomistic simulations. Yang et al.[29] used density functional theory (DFT) to compute the formation free energies of Ca-silicate ion pairs, and Li et al. explored the energy landscape of Ca coordination in $[Ca(H_2O)_x(SiO_2(OH)_2)]$ complexes by ab-initio metadynamics[30]. Regarding PPs, Manzano et al. proposed a C-S-H PP after cleaving a "basic building block" from tobermorite and jennite crystals, which was relaxed with ab-initio calculations[31]. They suggested that these PPs could aggregate differently depending on the $Ca^{2+}$ content in solution. Finally, a recent study has explored a two-step route for the C-S-H nucleation[32]. First, portlandite PPs are formed, and second, the reaction of silicate dimers with the portlandite PPs will induce a topochemical phase transition to form C-S-H monolayers.

In this work, we will investigate the structure and stability of C-S-H primary particles using atomistic simulation methods, specifically DFT and evolutionary algorithms (EA). Our goal is to determine the structure of energetically accessible small clusters (with the composition $nCaO + nSiO_2 + xH_2O$ where $n = 1, 2, 4$) that might appear during the initial stages of C-S-H nucleation. Then, we compare the structural resemblances of the PPs with the bulk C-S-H and study their aggregation kinetics with MD. Based on our simulations and the available experimental data, we suggest a formation pathway for the C-S-H nucleation.

## Results

In the following subsections, we will show the results for C-S-H clusters of different sizes and water contents. The general composition is $nCaO + nSiO_2 + xH_2O$ with $n = 1, 2, 4$ and $x$ variable from size to size. Following the contracted nomenclature used in cement chemistry, the clusters will be named as $C_nS_nH_x$ where C stands for CaO, S stands for $SiO_2$ and H stands for $H_2O$. Although the average value of Ca/Si ratio for C-S-H in cement is ~1.7, the Ca/Si ratio for the EA was chosen to be 1 and kept constant for all the clusters, which deserves an explanation. For the smaller clusters, $CSH_x$, there is experimental[33-35] and computational[36] evidence about the existence of Ca/Si = 1 complexes. The general belief is that the prevalent silicate species in the pore solution are $CaSiO_4H_2$ complexes (log $K = 4.6$), and the extra Ca will be in the form of a hydrated $Ca^{2+}$ ion. For larger clusters, there is no experimental information, so we assume as a working hypothesis that the $CSH_x$ complexes will aggregate and form dimeric species $(C_2S_2H_x)$. We cannot rule out that additional $Ca^{+2}$ ions will participate in the aggregation, forming dimeric clusters with Ca/Si > 1. However, given the structure of the $C_2S_2H_x$ that we will present in Discussion (2), we consider that the existence of Ca/Si = 1 dimers is plausible. Then, additional Ca should take part in the C-S-H structure in the aggregation stage. Consequently, for the MD simulations in Discussion (5) we added $Ca(OH)_2$ in the solution to increase the Ca/Si ratio to 1.5, a value closer to the actual C-S-H composition.

### $CSH_x$ complexes

First we seek for the most stable coordination complex of a silicate monomer with one Ca ion. Following the above criteria we chose $CSH_{10}$ as initial stoichiometry, which would corresponds to a neutral $CaSi(OH)_2O_2$ system plus 7 or 8 water molecules. It must be notice that the EA does not assume any charge state for the different units in the complex, it just keeps the atoms in the system constant. Therefore the EA can explore both complexes with Si-OH + Ca-OH or Si-O-Ca + water.

Figure 1 shows a sketch-map of the clusters found during the global minimization with energy lower than $E_0 + 150$ kJ mol$^{-1}$, $E_0$ being the energy of the lowest energy cluster. The sketch-map is a graphical representation in which clusters are represented by points, and the similarity between the clusters is depicted by the distance between the points. Closer points indicate a higher degree of structural similarity, while farther points represent a lower level of similarity. In Fig. 1a the color scale represents the energy difference, while in Fig. 1b–g the colors represent a mapping of several structural properties. The sketch-map shows 2 different regions. In region I the low energy complexes all contain a silicate tetrahedron bonded to a 6- or 7-fold coordinated Ca atom [Fig. 1c]. The silicon and calcium atoms are mostly bonded by a single Si-O-Ca bond, and 7 or 8 water molecules are found around the complex, with 4 or 5 filling the Ca hydration shell. In region II, the complexes include 5-fold coordinated silicate groups, with energies at least 1.35 kJ mol$^{-1}$ higher than those in region I. In this case, the silicon and calcium atoms are mostly bonded by two Si-O-Ca bonds [Fig. 1d]. The appearance of pentacoordinated silica is not a surprise as it is a common transient structure during condensation or hydrolysis reactions, especially in high pH environments[37].

It is important to notice that many complexes within accessible energies present 7-coordinated atoms [see Fig. 1c]. That is the coordination of Ca atoms in the C-S-H gel intralayer region[31,32], and it is common to use such a structural feature to characterize the early formation of C-S-H from Synchrotron X-ray powder diffraction (SXRPD)[38,39]. Our simulations indicate that the complexes may already display such coordination, and therefore they may interfere with the characterization of the solid phase. The same EA search has been made for stoichiometries with less water, $CSH_6$ and $CSH_2$, and are shown in the supplementary information (SI). Decreasing the water content does not change the formed complexes substantially: the best clusters also contain a silicate tetrahedron linked to the Ca atom, but the

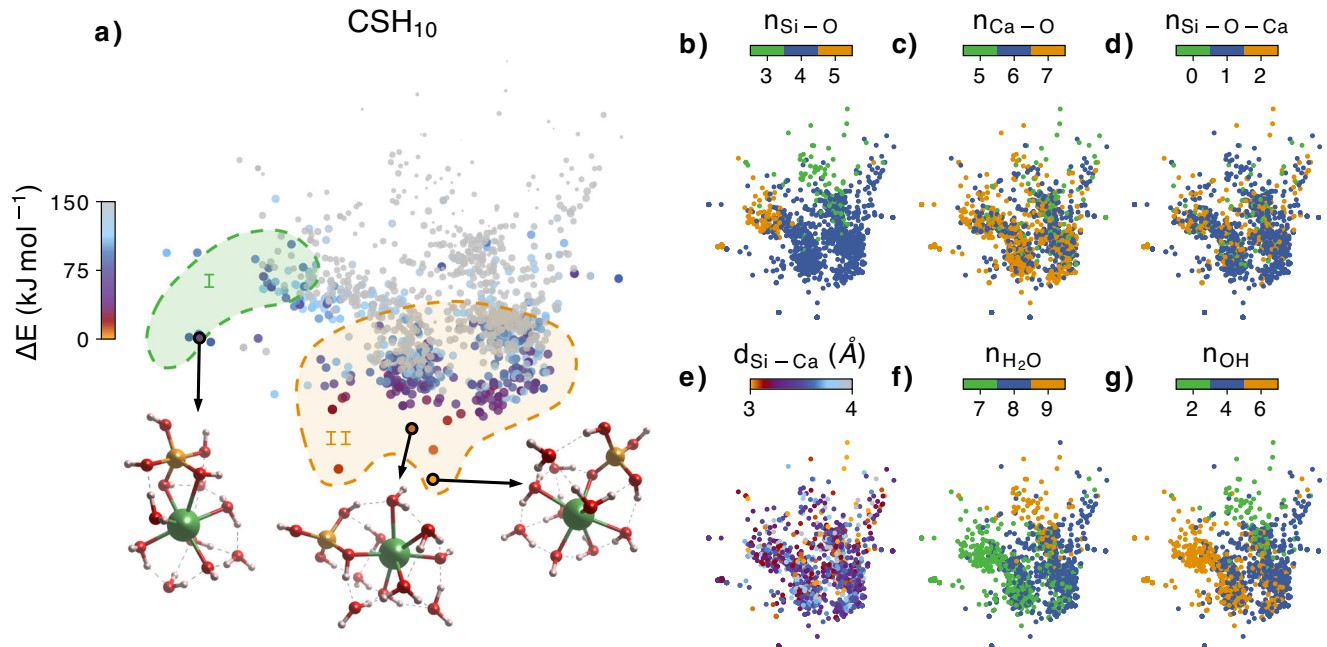

**Fig. 1 | Sketch-map representation of the CSH₁₀ clusters.** Each point represents a different primary particle (PP), and the distances between points represent the structural similarity between the PPs. The closer the points are, the more similar their structures are and vice versa. The colormap in (**a**) represents the energy difference $\Delta E$ with respect to the lowest energy PP. The atomic structure of the best PPs are also shown, with Ca atoms represented in green, Si atoms in orange, O atoms in red and H atoms in white. The regions I and II each represent a group of low energy PPs with similar structural properties. **b**–**g** Collection of structural properties of the clusters color mapped into the same Sketch-map. $n_{Si-O}$, $n_{Ca-O}$ and $n_{Si-O-Ca}$ represent respectively the number of Si-O, Ca-O and Si-O-Ca bonds. $d_{Si-Ca}$ represents the distance between the Si and Ca atom in the PP. $n_{H_2O}$ and $n_{OH}$ represent the number of water molecules and hydroxil groups in the PP. Source data are provided as a Source Data file.

coordination of the Ca atoms decreases due to the lack of water molecules.

Overall, the most stable complexes found by EA are similar to those reported before by atomistic simulation methods[29,30,36], yet direct comparisons are not possible. For instance, Li et al. studied the energy landscape of Ca-coordination in a $CaSiO_4H_2$ neutral cluster, but they did not consider the 7-fold coordination that we find predominant. Galmarini et al. reported the free energy landscape for the aggregation of a $SiO_2(OH)_2^{2-}$ and $Ca^{2+}$. The complex formation is favorable, with an Ca-Si equilibrium distance of 3.6 Å, in excellent agreement with the distance of the most stable clusters found by EA, [Fig. 1e]. However, their conformation is fixed and the protonation state is not the same as the the predicted by EA.

### $C_2S_2H_x$ clusters

The next step in the C-S-H nucleation would be the formation of primary particles by the interaction (merging) of complexes in solution. Therefore, we have explored the most stable PPs for the stoichiometry $C_2S_2H_{20}$. At this stage is clear the potential of EA to unravel the structure of the PPs: the complexes from "CSHx complexes" have a relatively simple structure that can be guessed by chemical intuition, but for larger systems, the task could be more complicated. Thanks to EA, we do not need to assume any initial structure or feature, and the global optimization will lead us to the lowest energy structures.

Two sets of low energy structures can be distinguished in the sketch-map in Fig. 2a for the $C_2S_2H_{20}$: region I, where the most stable candidates are found, with silicate dimmers, and region II with isolated silicate monomers [blue and green respectively in Fig. 2d]. The energy difference between the best candidate from region I and the best candidate from region II is just 6.7 kJ mol⁻¹, which suggests that both PPs could cohexist at room temperature. The best structures of region I contain mostly 6-fold coordinated Ca atoms, although some 5- and 7-fold coordinated Ca are also present, and the silicate groups are

mainly tetrahedral. In region II, the average Ca-O coordination is slightly larger, ranging from 5.5 to 6.5, and although the silicate groups mostly contain 4-fold coordination, some 5-fold coordinated silicates are present. A constant characteristic for low energy structures from both regions is that the Ca atoms are linked (by oxygen atoms and/or hydroxyl groups) and never separated.

We performed simulations decreasing the water content of the $C_2S_2H_x$ cluster from $x = 20$ to $x = 12$, 4 (see the SI). As expected, the coordination of the Ca ions decreases, but the main structure of the silicate arrangement remains, with the two Ca atoms linked to each other and to silicate monomers or dimmers. We found that the number of low-energy structures containing monomers increases as the water content decreases, probably to maximize the coordination of Ca atoms.

### $C_4S_4H_x$ clusters

We went one step further simulating clusters with the stoichiometry $C_4S_4H_x$, which would correspond to the aggregation of 2 $C_2S_2H_x$ PPs. For the $C_4S_4H_8$ PPs we divided the similarity map into two regions. Region I comprises low energy PPs with heterogeneous characteristics in terms of silicate chain network, number of water molecules and OH groups, as shown in Fig. 3b–d. Nevertheless, the lowest energy structures within region I share similar characteristics: they are formed by a silicate monomer and a silicate trimer, and their gyration radius is <0.3 nm, at the lower end of the range for all the structures. Therefore, a more compact structure is energetically favorable. Region II contains clusters with a silicate dimer and two monomers. The energy difference between the best structures in region I and II is 11.1 kJ mol⁻¹, a non-negligible value that points to the higher stability of the longer silicate chains in region I. It is specially interesting the emergence of silicate monomers and trimers within the most favorable clusters. It is usually believed that trimers do not appear at any stage of C-S-H formation, and monomers disappear soon after hydration starts[40,41]. However,

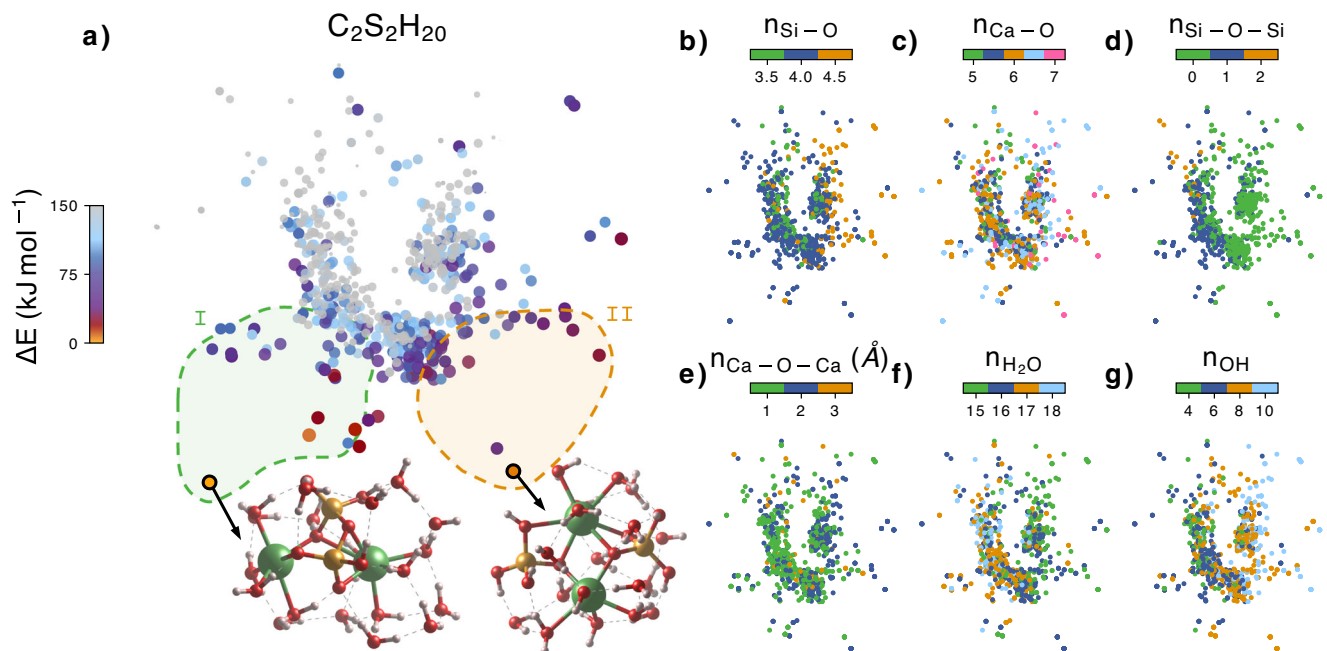

**Fig. 2 | Sketch-map representation of the $C_2S_2H_{10}$ clusters.** Each point represents a different primary particle (PP), and the distances between points represent the structural similarity between the PPs. The closer the points are, the more similar their structures are and vice versa. The colormap in (**a**) represents the energy difference $\Delta E$ with respect to the lowest energy PP. The atomic structure of the best PPs are also shown, with Ca atoms represented in green, Si atoms in orange, O atoms in red and H atoms in white. The regions I and II each represent a group of low energy PPs with similar structural properties. **b**–**g** Collection of structural properties of the clusters color mapped into the same Sketch-map. $n_{Si-O}$, $n_{Ca-O}$, $n_{Si-O-Si}$ and $n_{Ca-O-Ca}$ represent respectively the number of Si-O, Ca-O, Si-O-Si and Ca-O-Ca bonds. $n_{H_2O}$ and $n_{OH}$ represent the number of water molecules and hydroxil groups in the PP. Source data are provided as a Source Data file.

recent works report the presence of $Q_{0,1,2,3}$ NMR signals for synthetic C-S-H at very early nucleation times that change over time until only $Q_{1,2}$ remain[9]. That suggests a wide population of silicate oligomer structures in the amorphous globules that evolve during crystallization to form the linear chains following the well known 2n-1 length rule[41].

For this size, we also include the less hydrated PPs with $C_4S_4H_2$ stoichiometry [Fig. 3e], because the most stable cluster structure is of special interest. Due to the lower amount of water molecules, the obtained clusters do not cover a large configurational space, and they are very homogeneous. We only distinguish one region, with clusters made of 2 silicate dimers linked by Ca atoms, and no water. The lowest energy structures are symmetrical, and the Ca atoms form a central layer with the silicate dimers at the top and bottom, which resembles the atomic arrangement in tobermorite. In fact, the clusters have a very similar structure as the one reported by Manzano et al. in 2007[31]. In that study, a common structural feature was found in tobermorite and jennite crystals, which are considered perfect versions of the C-S-H in cement, and was isolated to build a cluster. The cluster was relaxed in vacuum using ab-initio simulations, and it was proposed as the smallest building block of C-S-H. The exact same cluster isolated and relaxed from the crystals is also found by EA as one of the lowest energy structures, with an energy of just 5.1 kJ mol[−1] larger than the global minimum obtained by EA. The only structural difference between the lowest energy cluster and the one reported in ref. 31 is the orientation of the silicate dimers, perpendicular in the former and parallel in the latter. The fact that EA predict a low-energy structure with the same atomic arrangement as tobermorite strongly suggests that the $C_4S_4H_2$ cluster could indeed be the C-S-H basic building block.

### Aggregation of the primary particles

After determining the lowest energy clusters for the different sizes, we explored different steps of the possible C-S-H aggregation pathway. First, we computed the sequential enthalpy of formation of the complexes/PPs formed by merging smaller structures. For each size, the

lowest energy cluster was relaxed using DFT including an implicit solvent effect.

The first step is the formation of $CSH_x$ complexes from the ions in solution. The ions have been selected based on the predominant individual speciation states of Ca and silicic acid in high-pH environments. According to Refs. 42,43, Ca is present mainly as $Ca^{2+}$ with a minor presence of $Ca(OH)^+$, and silicates mainly as $H_3SiO_4^-$ and $H_2SiO_4^{2-}$ (see SI for details). To better reproduce the solvent effect, a variable number of water molecules were explicitly added to the species in solution based on the estimation by Yang and White[29]. All the enthalpies of formation and reaction stoichiometries for the complex formation are shown in the SI. Figure 4a shows the most favorable ones relevant to our discussion, which corresponds to the pathways starting from the ionic species with the highest charges $Ca^{2+}(H_2O)_6$ and $H_2SiO_4^{-2}(H_2O)_6$ as reactants. The formation of the $CSH_{10}$ complex from the species in solution is very exothermic, indicating a considerable stabilization of the ions. It must be noted that despite the initial species in solution being charged, the complex is neutral, with a $H_3SiO_4^-$ silicate sharing its deprotonated oxygen atom with the Ca atom, which completes its hydration sphere with five water molecules and a hydroxyl group. To better understand the speciation at high pH, charged complexes and PPs will be considered in future works. Nevertheless, we can suggest that the formation of the $CSH_{10}$ complex is very favorable and it can be a dominant complex in cement pore solution, in agreement with previous simulations[36] and experiments[33–35]

The second step is the formation of the $C_2S_2H_{20}$ PPs by merging two complexes. Based on the experimental data, the dimeric PPs could be of key importance in C-S-H nucleation. During the hydration of tricalcium silicate, dimers are formed after ~1 h and are the dominant silicate species at all stages[40]. According to our simulations, the formation of $C_2S_2H_{20}$ from the complexes or from species in solution is also favorable. Based on the energetics and kinetics, we can expect that two complexes would merge into a $C_2S_2H_{20}$-II, with two independent

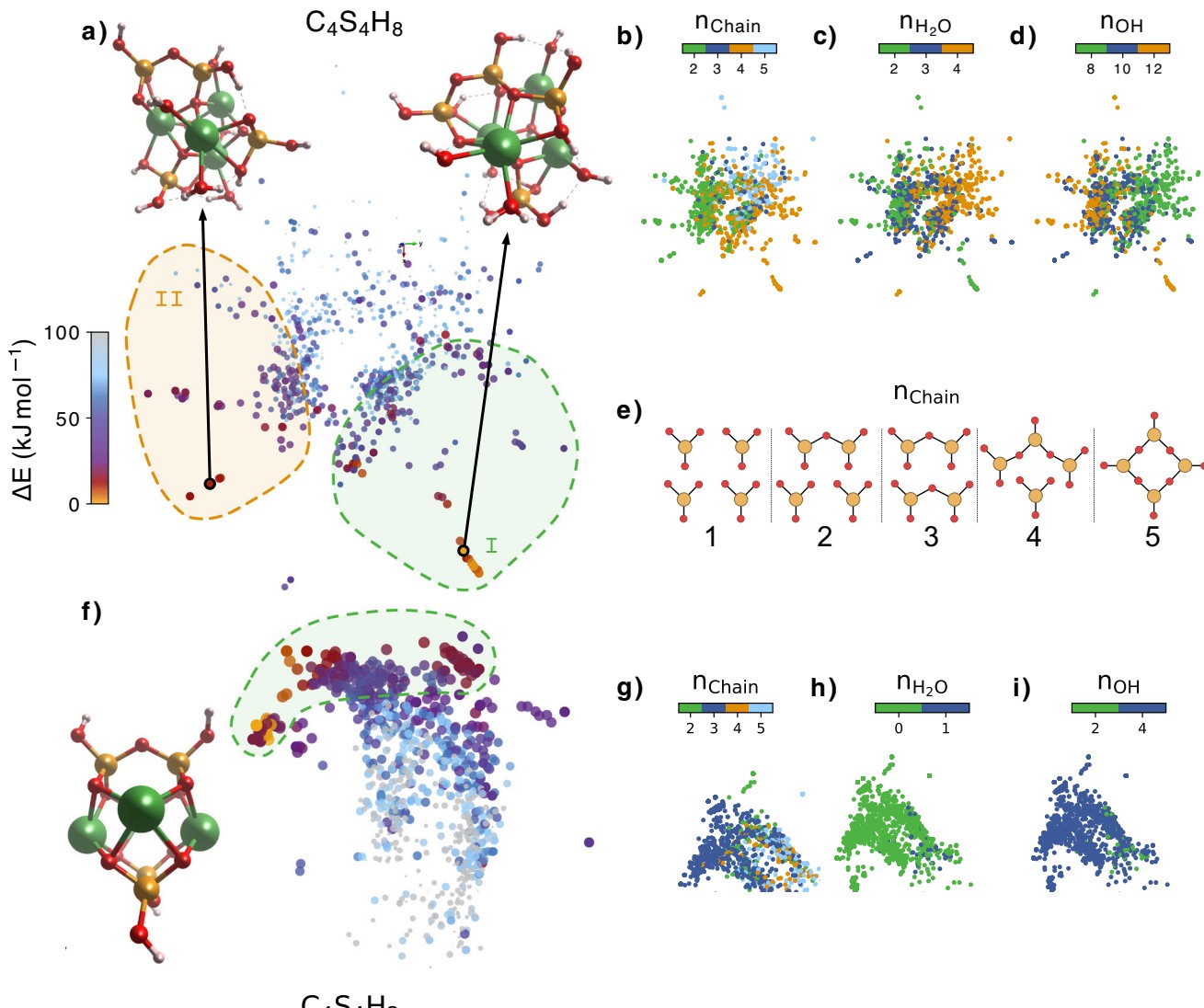

**Fig. 3 | Sketch-map representation of the C₄S₄H₈ and C₄H₄S₂ clusters.** Each point represents a different primary particle (PP), and the distances between points represent the structural similarity between the PPs. The closer the points are, the more similar their structures are and vice versa. The colormap in (**a**–**f**) represents the energy difference ΔE with respect to the lowest energy PP for each stoichiometry. The atomic structure of the best PPs are also shown, with Ca atoms represented in green, Si atoms in orange, O atoms in red and H atoms in white. The regions I and II each represent a group of low energy PPs with similar structural properties. **b**–**d** Collection of structural properties of the clusters color mapped into the same Sketch-map for C₄S₄H₈ PPs. **e** The different silicate arrangements found in the PPs. **g**–**i** Collection of structural properties of the clusters color mapped into the same Sketch-map for C₄S₄H₂ PPs. $n_{chain}$ shows the silicate arrangement as shown in (**e**). $n_{H_2O}$ and $n_{OH}$ represent the number of water molecules and hydroxil groups in the PP. Source data are provided as a Source Data file.

silicate monomers. Then, an intra-cluster condensation reaction would take place to form the dimer. The free energy of the dimeric $C_2S_2H_{20}$-I cluster is slightly larger than that of the monomeric cluster, and therefore the total free energy gain must be attributed to an increase of the entropy due to the released water molecule, as suggested before[16,18,44]. The $C_2S_2H_{20}$-II structure is noteworthy: the Ca atoms are close to each other and opposite to the siloxane bond between the silicates. As shown in Fig. 4b, such an arrangement can be found in tobermorite crystals, as well as in C-S-H and jennite crystals (see Fig. S17)[5]. This suggests that the silicate dimerization stage is the critical step that directs the formation of the C-S-H gel towards tobermorite-like lamellar structures.

To analyze the aggregation dynamics of the $C_2S_2H_{20}$ PPs we performed four MD simulations with different (and random) initial distribution of eight $C_2S_2H_{20}$ PPs in water. To simulate the experimental composition of the amorphous spheroids[6] and move closer to the experimental Ca/Si ratio of the C-S-H in cement pastes eight Ca(OH)₂

groups were added to the solution. After an equilibration period in the NPT ensemble, the silicate density in the simulation box was 0.38 ± 0.02, nearly 19 times lower than in the C-S-H and matching perfectly with the experimental estimation[6]. The mass density was $\rho = 1.02 \pm 0.02$ g cm⁻³, close to the bulk water density, which indicates a liquid-like nature of the amorphous spheroids. We observe that the $C_2S_2H_{20}$ PPs do not dissociate, i.e., the calcium atoms remain attached to the silicate dimers, so the dimeric PPs are stable.

The PPs and Ca(OH)₂ complexes in solution are highly dynamic and tend to form aggregates that grow over time, even within the limited simulation time (15 ns). The aggregates were defined using a distance cutoff between atoms of 2.7 Å, which accounts for the typical Ca-O distance[45]. An example of an aggregate structure comprising four PPs and 2 Ca(OH)₂ complexes is depicted in Fig. 4c. In Fig. 4e we show the size of the larger aggregate $S$ as a function of time, averaged over four simulations and normalized by the size of one $C_2S_2H_{20}$ PPs $S_0$. It can be seen the PPs aggregate very fast in short times, and they are

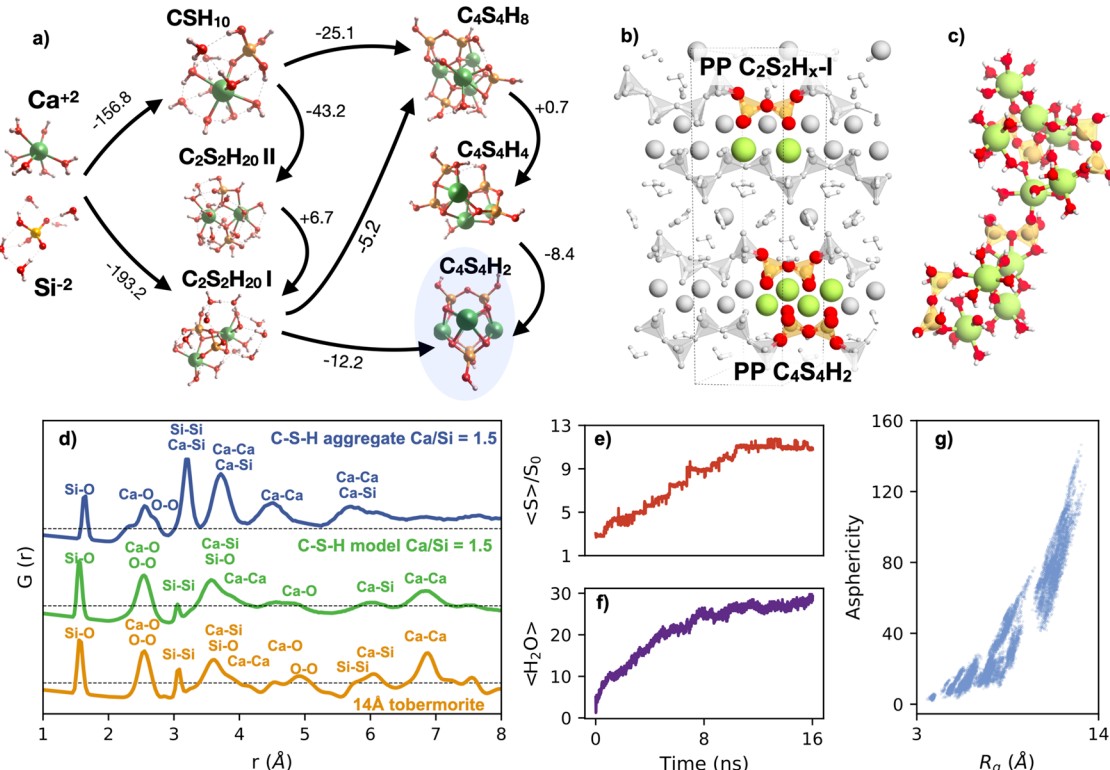

**Fig. 4 | Aggregation of the C-S-H complexes and primary particles. a** Free energies of the primary particles (PP) aggregation for several formation paths. The values are given in kJ mol⁻¹. **b** Representation of the Plombierite (tobermorite 14 Å) atomic structure. The two highlighted features are comparable to the atomic arrangement of the $C_2S_2H_x$ -I and the $C_4S_4H_2$ prymary particles (PP). **c** Example of an aggregate formed from $C_2S_2H_{20}$ clusters in solution after 10ns of MD simulation. **d** Computed pair distribution functions G(r) for the largest aggregate over the last 5ns of simulation, and its comparison with the computed G(r) of a bulk C-S-H model with Ca/Si = 1.5[66] and tobermorite 14 Å. **e** Evolution of the averaged aggregate size < S > over time. The size is defined in terms of number of atoms that for the aggregate normalized by the number of atoms in a primary particle (PP) $S_0$. **f** Average water loss < H₂O > from the aggregates over time. **g** Asphericity of the aggregates as a function of the gyration radius $R_g$. Source data are provided as a Source Data file.

dynamic, with PPs moving back and forward from the cut-off distance. Figure 4e shows the asphericity of the clusters as a function of the gyration radius. Clearly, larger clusters tend to be more elongated, in agreement with the structure of the DOLLOPs and rod-like particles proposed as primary particles for $CaCO_3$ and $CaSO_4$ nucleation[14,44]. The elongated shapes maximize the Ca-water coordination, which might stabilize these aggregates until they assemble into larger morphologies. A similar mechanism has been proposed for the nucleation of gypsum[14], with primary species that form aggregates without any coherent diffraction signal despite their large size, which suggests a disorder arrangement. The solvation water that we observe in this work may be the reason for the misalignment and disorder between the aggregates.

To analyse the internal structure of the aggregates we have computed the pair distribution function G(r) for the last 5ns of simulation, when the aggregate size has stabilized to its maximum value, and compare it with that of tobermorite 14 Å and a bulk C-S-H model. The diffraction peaks were identified by examining their distinct components, as displayed in the SI, and the water contribution was eliminated by subtracting the weighted G(r) of bulk water. Below 3 Å the peaks correspond to the first Ca-O, Si-O and O-O neighbors, and therefore are similar for the aggregates and the bulk systems. Above 3 Å the peaks for the aggregate are dominated by the contribution of the heavy atoms Ca and Si. The Ca-Ca, Si-Si, and Ca-Si signals are considerably stronger than in C-S-H and tobermorite. Conversely, the signals involving O are negligible, and the correlation is lost after 6 Å due to the shape of the particles. The lack of defined peaks for the X-O pairs beyond the bonded atoms suggest a considerable disorder: the short-range coordination of metal cations is the expected one, with

$SiO_4$ silicate tetrahedra and $CaO_6$ heptahedra, but the orientation and arrangement of these species does not have any medium-range order. Therefore, we cannot state that the aggregates present a C-S-H-like structure yet.

The last step in the formation path involves the formation of $C_4S_4H_x$ PPs. Based on the fast aggregation of $C_2S_2H_{20}$ in MD simulations, and the fact that EA predict clusters with silicate trimers as the most favorable ones, we suggest that $C_4S_4H_x$ clusters may not present in solution. However, the $C_4S_4H_2$ structures can be seen as C-S-H basic building blocks, as shown in Fig. 4b, so we propose that they will be formed by progressive dehydration within the large $C_2S_2H_{20}$ aggregates. In fact, dehydration has been suggested to be a key step in $CaCO_3$ nucleation, as releasing water leads to an increase of the entropy, reducing the free energy of the nuclei[16,18,44]. Our DFT simulations of isolated clusters indicate that dehydration is slightly favorable. More interesting is the evolution of the $C_2S_2H_{20}$ aggregates in solution. As can be seen in Fig. 4g, the mean number of water molecules attached to the aggregates decreases with time. The decrease is due to the aggregation itself, since the PPs lose water molecules from their hydration shells, and also due to rearrangements within clusters, including condensation reactions between the $C_2S_2H_{20}$ and $Ca(OH)_2$. Despite only observing the initial steps of dehydration, we suggest that $C_4S_4H_2$ clusters will be progressively formed within the aggregates leading to crystallization.

## Discussion

Based on all the obtained results we can propose a possible C-S-H nucleation path. For that we have used as a starting point the three stages observed in turbidity experiments as a function of time[6] (see

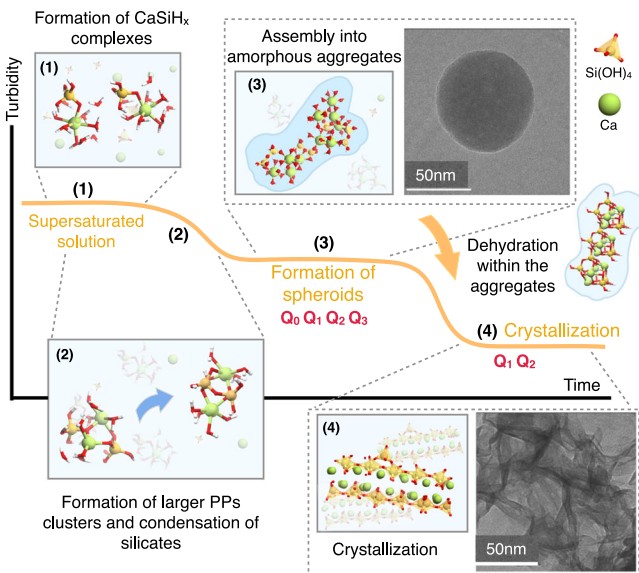

**Fig. 5 | Schematic representation of the C-S-H nucleation pathways.** The shape of the turbidity vs time defines 3 stages stages in the nucleation process (based on ref. 6). The experimental TEM image in step 3 shows an amorphous C-S-H globule, and the TEM image in step 4 the formation of C-S-H layers. The experimental images have been adapted from ref. 26, Pages No. 33-39, Copyright 2018, with permission from Elsevier. The $Q_n$ signals in steps (3) and (4) are based on ref. 9, and represent the connectivity of the silicate groups.

Fig. 5). The suggested nucleation path can be described in 4 steps, the two initial ones within the first stage of nucleation:

1. The main species in the supersaturated solution, $Ca^{2+}$ and $SiO_4^{2-}$, form $Ca(OH)\text{-}O\text{-}Si(OH)_3$ complexes. Energetically, the reaction is very favorable, and previous investigations reported a barrierless aggregation[36], so the formation of the complexes is expected to be fast.
2. The complexes merge into $C_2S_2H_{20}$ primary particles (PPs). The silicate dimers are formed via condensation reaction within the PP.
3. The $C_2S_2H_{20}$ PPs are very dynamic and rapidly aggregate into large clusters. These large aggregates tend to be elongated and they are highly solvated (possibly stabilized) by water. Groups of clusters more or less entangled would form the experimentally observed amorphous globules. If the experimental silicate density is correct[6], the density of the globules should be close the bulk water value, $\rho \approx 1.02\ \text{g cm}^{-3}$, so the globules could be seen as liquid-like in nature. In this stage NMR experiments indicate that the silicate oligomers may include species different from dimers[8,9] which we have not explored in this work, yet EA predicts favorable clusters with silicate monomers and trimers.
4. The large aggregates slowly dehydrate, and $C_4S_4H_2$ structures are formed within them. These $C_4S_4H_2$ structures present a tobermorite-like structure and can be seen as the basic building block for the C-S-H formation. Finally, a slow process of dehydration and rearrangement of the $C_4S_4H_2$ within the aggregates will result in the C-S-H layers. If present, branched silicate oligomers should de-polimerise during this crystallization stage to form linear chains.

Overall, we have used atomistic simulation methods to investigate the structure and formation of C-S-H complexes and PPs, and then discuss a possible C-S-H nucleation pathway based on the simulations and evidence from the literature. This work is the first computational step in the challenging path toward understanding and controlling C-S-H nucleation, and have significant implications for advancing in the design and optimization of cementitious materials. A deeper understanding of the nucleation mechanisms can aid in tailoring additives to regulate it, modifying the rates and mechanisms, and therefore controlling the rehology, the pore structure, the strength, or the durability. Nevertheless, we must keep in mind that the actual nucleation path may be different depending on the C-S-H formation process (synthetic, from $C_3S$ dissolution, from cement hydration) and the presence of guest ions, the co-nucleation of other phases, etc. In future work, the role of these variables should be explored, as well as seek for a more detailed and quantitative description of the aggregate formation and the dehydration/crystallization stage.

## Methods

The prediction of the most stable clusters was done using evolutionary algorithms as implemented in the USPEX code (v.10.3)[46–48]. For each system, a first generation of 500 clusters were created and relaxed using the ReaxFF force field[49] with the Ca/Si/O/H set of parameters from Refs. [45,50,51]. From the first generation, the best 25 clusters were selected to start the evolutionary algorithm. In each generation, the clusters were relaxed in two steps to accelerate the calculations, first using empirical potentials (ReaxFF) and then DFT. The energy minimization with empirical potentials was done with GULP(4.2.0)[52] using the conjugate gradient and the ReaxFF force field[45,50,51]. The DFT minimizations were also done with conjugate gradients using the SIESTA code (4.1)[53]. The electron exchange-correlation has been taken into account at the generalized gradient approximation (GGA) using the Perdew-Burke-Enrzerhof (PBE) functional[54]. Grimme dispersion correction was used to include van der Walls interactions[55]. A double-$\zeta$ polarized (DZP) basis set was used for the pseudo-atomic orbitals. The best structures were further minimized using the Polarizable Continuum Model (PCM) to simulate the high pH environment, using a dielectric constant of $\epsilon = 56$, which corresponds to NaOH solution with a pH of around 14[56]. For that, the Gaussian09 code was use[57], with the PBE XC functional, a 6-311+G(d) basis set[58] and the empirical dispersion by Grimme[55].

To compute the structural similarity between the clusters, the smooth overlap of atomic positions (SOAP) was used[59]. The Ca, Si and O atoms were used as centers of the SOAP descriptor, with a cut-off of 3.5 Å. 16 radial functions and 12 angular functions were used to expand the descriptor and the best-match structural kernel was used to compute the final structural distance between structures[60]. Then, the Sketch-map method was used to reduce the similarity matrix to a 2D representation of the explored phase space[61,62]. These 2D representation maps together with the atomic structure of the clusters are included as .json files in the supplementary material, ready for interactive visualization in the chemiscope online tool[63] (for more information see SI).

The molecular dynamics (MD) simulations were done using LAMMPS (21-Jan-2020)[64] and the ReaxFF force field. Four independent simulations were done starting from different configurations generated by random packing of 8 $C_2S_2H_{20}$ clusters and 8 $Ca(OH)_2$ molecules with 700 $H_2O$ molecules in a 7 nm side box. After 200 ps for density equilibration in the NPT ensemble, the simulations were run for 15 ns in the NPT ensemble, with a 0.2 fs timestep and using the Verlet integrator.

## Data availability

Sample input files of the USPEX, SIESTA, GAUSSIAN, and LAMMPS codes are given in the SI, reaxFF force-field parameter can be found in a repository[65], json files with the results from the EA ready for interactive visualization in the chemiscope online tool[63], and the output data of all the structures are provided as a supplementary material. Source data are provided with this paper.

## Code availability

All the software used in this work is open source. No specific software was developed for this work.

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

## Acknowledgements
This work, was supported by the "Departamento de Educación, Política Lingüística y Cultura del Gobierno Vasco" (Grant No. IT1458-22), the "Ministerio de Ciencia e Innovación" (PID2019-106644GB-I00 and TED2021-130860B-I00), the University of the Basque Country UPV/EHU (Colab22/06) and the Laboratory for Trans-border Cooperation "Aquitaine-Euskadi Network in Green Concrete and Cement-based Materials" (LTC-Green Concrete). The authors thank for technical and human support provided by SGIker (UPV/EHU/ ERDF, EU). X.M.A. acknowledges the financial support from the University of the Basque Country UPV/EHU (PIF17/118), and J.L.-Z. the financial support from the Basque Country Government (PRE_2019_1_0025).

## Author contributions
X.M.A., I.E. and H.M. conceived the project. X.M.A. performed the simulations. X.M.A., J.L.-Z. and H.M. analysed the data. X.M.A. wrote the manuscript first version, and all authors reviewed the manuscript. I.E. and H.M. supervised the research.

## Competing interests
The authors declare no competing interests
