## [Peer Review File · Nature Communications]

Multi-step nucleation pathway of C-S-H during cement hydration from atomistic simulationsREVIEWER COMMENTS

Reviewer #1 (Remarks to the Author):

In this manuscript, Aretxabaleta et al present a vast collection of numerical data from free energy calculations of C-S-H prenucleation clusters (PNCs), specifically for Ca/Si ratio 1 for compositions $n=1, 2, 4$ and x amount of water molecules. The energetically favoured structures are identified and compared to bulk structure of tobermorite and a nucleation path is proposed.

C-S-H has various structures that depend on the Ca/Si ratio and it is mostly amorphous. The crystalline structure of tobermorite is often found in hardened cement paste but it is not evident that PNCs have to resemble tobermorite. Some recent experiments of Bae et al, *Materials* 2016, 9(12), 976 have reported Ca containing nanoparticles as precursors of C-S-H. Why not investigating a Ca/Si ratio around 1.7 that is the most common found in cement paste? I suppose the results will be very different. How can we know from this work that the PNCs identified here are most relevant ones for cement paste?

The data are presented via sketch-maps which is an elegant way to visualise such a large amount of data. In the SI, input files for the simulations are provided. Given the plethora of computational codes used and the type of data provided in this article, it is impossible to judge the soundness of the simulations, except if one repeats the simulations. The authors could have include more typical simulation plot in the SI.

This article has noteworthy results for a specialised audience. The lack of comparison with experimental data and the specific choice of Ca/Si=1 make it difficult to convince that these PNCs will be relevant to the complex dissolution and nucleation processed in cement. Therefore, I wouldn't recommend publication to broad audience journal.

Some minor comments:

- in the figures nomenclature is used that hasn't defined in the text or caption.
- few references are not in the correct format, ref. 3 and 4 are the same.

Reviewer #2 (Remarks to the Author):

This is an impressive work employing a whole hierarchy of simulation methodologies in a clever way, in order to address the puzzles of the formation of cement. The work is important and appears to be correct. I recommend to publish it, with optional minor revision.

Some comments:

1. I recommend to pay more attention to describing clearly the implications of the results. Where can we expect complex nucleation mechanisms? Can we use present results to improve cement?
2. I spotted a typo on the bottom of p. 4 - should be $H_3SiO_4^-$, not $H_3SiO_8^-$.
3. The "sketch maps" were first developed in this paper, which would be correct to mention: Oganov A.R., Valle M. (2009). How to quantify energy landscapes of solids. *J. Chem. Phys.* 130, 104504.

None of these minor comments decrease the quality of this work, which, as I wrote, should be accepted after minor revision.

Artem R. Oganov

Reviewer #3 (Remarks to the Author):

This study examines the initial stages of the formation of calcium silicate hydrate (CSH), the primary binding component of cement. The authors employed various atomic simulation techniques to better understand the clustering that occurs during the prenucleation stages of CSH formation. The study offers new perspectives on this topic, which align well with previously reported experimental and computational findings. However, it would be useful for the authors to provide a comprehensive illustration that integrates their results with those already reported in previous literature.

The manuscript is well-written and, given the significance of cement, will likely attract a wide range of readers. However, before it is published, there are a few aspects of the manuscript that could be enhanced, as outlined in the comments below.

Detailed comments:

Abstract.

L7, Recent evidences indicate that the C-S-H nucleation is non-classical, yet detailed information on the nucleation steps is still missing.

>> be more specific about which 'detailed information' is still missing. (both in the abstract and introduction)

Introduction

L30. In non-classical nucleation (NCN), the exact number of steps and nucleation paths are system dependent, yet there are generalities. First, the existence of thermodynamically stable pre-nucleation clusters (PNCs).

>> Rather than a generality, I would say PNCs are a common feature in nonclassical nucleation pathways, but definitely not a necessity .

>> The authors should provide a definition of "PNCs", since formally these are clusters which form in solution conditions undersaturated with respect to the final phase (e.g. Gebauer et al, Science, 2008). In this respect it is not clear if the clusters discussed in this studies are PNC's sensu stricto (i.e. forming already at concentrations below the solubility of the final phase) or not?

L31. The PNCs are small, formally do not have a phase interface, and should be considered solutes rather than solids.

>> define "small"

L40. Direct experimental observation of PNCs structure and dynamics is extremely challenging due to the time and size scales. Atomistic simulations have identified the formation of dynamically ordered liquid-like oxyanion polymers (DOLLOPs) formed by ion pairs in Ca-phosphate and Ca-carbonate. However, for barite (BaSO₄) compact clusters resembling crystalline motifs have been reported.

>> Experimental observations of PNCs have been reported for calcium sulfate, where the formation and evolution of PNCs to the final phase has been monitored in situ (Stawski et al., Formation of calcium sulfate through the aggregation of sub-3 nanometre primary species. Nat. Commun., 2016, 7, 11177) and the a tentative structure of the PNCs, based on scattering data and MD, has been put forward (T. M. Stawski et al., The Structure of CaSO₄ Nanorods: The Precursor of Gypsum. J. Phys. Chem. C 123, 23151, 2019). I would be meaningful to includes these result into this part of the introduction.

L47. In the particular case of synthetic C-S-H, dynamics light scattering (DLS), small-angle X-ray scattering (SAXS), and cryo-TEM have shown the formation of "amorphous solid spheroids" of about 60 nm that reorganize into C-S-H platelets after around 5h.

>> It would probably make sense to specify the conditions of these precipitation experiments (e.g. diluted solution, direct mixing of solutions,..).

Results

Therefore, the Ca/Si ratio is 1 and kept constant for all the clusters.

>> Indicate that this ratio is different from the typical Ca/Si ratio of CSH in cement pastes and provide a brief state of why this ratio was selected (could be added to Mat&Met).

>> Perhaps the authors could speculate on how a ratio higher than 1 would influence the prenucleation clusters structure?

Discussion

L286: in the context of non-classical nucleation theory.

>> At present there is not yet a formal theory for non-classical nucleation. More correct would be to say: non-classical nucleation models.

(as far as I know the only theoretical framework dealing with non-nucleation can be found in the works of Lutsko: e.g. Science advances 5 (4), eaav7399; New Journal of Physics 20 (8), 083019)

L333: two typos ...in the CSH... five of MD...

L356: The aggregates are strongly solvated by water molecules, so they tend to be elongated. Figure 4(e) shows the asphericity of the clusters as a function of the gyration radius. Clearly, larger clusters tend to be more elongated, resembling the DOLLOPs found during CaCO₃ prenucleation.

>> Any reason why aggregates tend to elongate due to solvation?

>> SAXS measurements have revealed that in the case of CaSO₄ clusters are also elongated (Stawski et al., Nat. Commun., 2016, 7, 11177).

L381: Despite only observing the initial steps^[1] of hydration, we suggest that C₄S₄H₂ clusters will be progressively formed within the aggregates leading to crystallization, following the experimental steps found experimentally by Krautwurst et al. 7

>> it would be insightful if the authors could include a schematic figure of the nucleation pathway of CSH, combining their results with those previously reported. In addition, this could serve to detail what information is still missing to establish a holistic view of CSH nucleation.

Alexander.

Reviewer #4 (Remarks to the Author):

General comments

A very interesting paper on a key topic for better understanding the earliest stages of the formation of Calcium Silicate Hydrate (C-S-H), the most abundant man-made material on earth. Despite many years of research there are still many unknowns with respect to C-S-H and this paper definitely contributes an interesting and original approach to the earliest moment of formation – pre-nucleation species from atomistic scale modelling.

I think the main limitation of the paper as submitted is the introduction. The authors do not set the scene adequately for what follows. My opinion is that the non-classical nucleation pathway suggested in reference 7 is a very circumspect proposal and is by no way proven or currently accepted in the community – it is just a possible interpretation of their data. Other interpretations could include a classical nucleation pathway with the first product being amorphous – as for amorphous calcium carbonate[1]. I think the authors should say that it is not clear from 7 that it is non-classical – albeit a possibility and then use their simulations to show in a first step how atomistic simulations can help contribute to unravelling the precipitation/nucleation pathway of C-S-H.

The paper is novel and I think breaks barriers that have not been broken before and promises to lead us towards a much better and atomistic scale view of C-S-H nucleation. However as stated above and discussed in the detailed comments below I feel the focus of the paper needs adjusting and many minor points need clarification. Therefore, my decision would be to accept but with major revisions.

Detailed comments

Line 6 I think C-S-H should be defined here as this is not a specialist cement journal – Calcium Silicate Hydrate

Line 6 abstract – evidences – should read - evidence

Line 6 abstract – I am not convinced that the nucleation is non-classical. It might be multi-step but there is plenty of evidence for a nucleation energy barrier, albeit possibly towards an amorphous first phase but will come back that in more detail below.

Lines 26-30 – so here we go ...the Non-classical nucleation misunderstandings; are the authors (and the references cited) really talking about non-classical nucleation theory or just indirect routes...which still have an energy barrier...see [1] SI for a very clear discussion and [2] for a clear schematic of such pathways. i.e. unknown activation energies (classical pathways) are not non-classical Many people induce the idea of a non-classical route when in fact what they mean is unknown formation pathway. Also, one has to be very clear about nucleation pathways and growth (atomistic, molecular, cluster, or aggregation (disordered and self-ordered)).

Understood that both ref 7 and ref 8 in the manuscript show that an initial amorphous globular phase transforms into the more recognisable sheet like structure of synthetic C-S-H but there is no strong evidence in either paper that this amorphous first phase does not have a nucleation barrier. In fact, there are many papers indicating that there is such a classical nucleation barrier and go on to very convincingly analyse the primary and secondary nucleation rates according to classical nucleation theory (CNT) [3-6]. There are many examples of classical nucleation routes that give an amorphous and/or extra hydrated first product which then transforms into the more thermodynamically stable phase later in the process as the saturation of the solution is modified ([1,7, 8]). I think many people have jumped on this non-classical nucleation band wagon for growth processes that do not follow the atomic growth unit pathway i.e. the mesoparticles made up of nanoparticles via aggregation steps [2] and often actually do not look at the nucleation process, which is a non-trivial process to investigate. I strongly disagree with the review cited by the authors Lee et al ref 10 – that having a non-crystalline first product is a non-classical nucleation process... the seminal work by Nielsen does not stipulate the product has to be crystalline [9, 1]. That is to say the first product does not have to be crystalline and the transformation from amorphous to crystalline can be classical...i.e. a classical nucleation barrier for the first crystalline seed then growth takes over [1,2,8] or non-classical-via aggregation or dehydration steps [2]. So, for the moment I see no evidence in the citations 7 and 8 which indicate that C-S-H nucleation is non-classical – it is just an interpretation. Also, in the applied cement systems used to make concrete etc the dominant mechanism is probably heterogeneous nucleation. I understand that sub-critical size nuclei – embryos can adsorb and become stable nuclei and thus promote growth, akin to the secondary nucleation process suggested for several systems where nucleation nonetheless can be described classical nucleation theory (CNT)[1,3,10]

I feel that the authors may in this paper be presenting the first real evidence of a non-classical nucleation pathway and thus they really need to get the introduction right.

Lines 61- 66 – I think the use of atomistic simulations for “pre-nucleation” species or clusters is a very important approach and will gain weight in coming years. However, I think that solution species – ion-pairs or hydrated ion pairs – are often in equilibrium with the simple ionic species that go to make them up. Such complexes in solution that can be the classical building block of the precipitate if this is the dominant(i.e. highest population) species in the solution.

Lines 69-72 – The two step route reported in reference 26 (the author’s own work) is really far from the non-classical route discussed here, so I am not too sure how relevant it is if the current work does not support their assumptions in ref 26. Despite the elegant calculations presented in ref 26 there is no evidence for this calcium hydroxide intermediate phase in any of the synthetic C-S-H synthesis papers that I am aware of, the references 7 and 8 certainly not, the globules contain Ca/Si. In fact, the C-S-H globular particles were first clearly seen in the work of Kumar et al ([11] – Fig S3) and they show clearly that the globules are not amorphous silica or amorphous calcium hydroxide with high resolution (3nm spot size Table S2). So I am very skeptical about this calcium hydroxide 2 step mechanism.

Line 111 is the last word in this line cluster ? it should read clusters..

Line 141 – I am not too sure what the normalisation to CS – CaO-SiO₂ really means wrt to the water that is then added...is this meant to be a stable anhydrous CaO -SiO₂ cluster or just the CaO and SiO₂

separate energies?

Lines 148 – 150 – it should be made clear that these are the species for separate calcium and silicate solutions...once mixed it is well accepted that the main silicate species becomes the neutral complex , $\text{CaSiO}_2(\text{OH})_2$, at pH's above 13 with the $\text{H}_2\text{SiO}_4^{-2}$ being an order of magnitude less in concentration.[12]

Line 164 – notice – should read – noted

Lines 183 -186- indeed the 7-fold coordination of Ca is a key structural feature seen long before these recent SXRPRD studies [13] – indeed it is a key motif in the Tobermorite structure, the basis of most C-S-H models [13, 14].). Generally, coordination of less than 6 is rarely seen for calcium in the C-S-H system [13].

Lines 190-197 the structures of the CSH complexes could also be compared to the much earlier work of Galmarini et al [12] where I believe $\text{CaSiO}_2(\text{OH})_2$ was first demonstrated to be a stable complex, in particular the Ca-Si distance compared to the results in the current manuscript. The complex is in equilibrium with the solution and its concentration (or activity) can be calculated under different solution conditions using relatively standard thermodynamic packages such as GEMS [15]. Although the paragraph is headed CSHx complexes ...the authors use the description cluster quite often in the paragraph and even in the Figure 1 caption...I think it is very important that this difference between expected complexes in thermodynamic equilibrium with the solution and the proposed clusters is maintained. That is to say, the clusters that may represent a stable thermodynamic state but are smaller than a perceived stable nucleus that would then grow and reduce its free energy when increasing its size according to the CNT.

Line 228 – 12, 4 – should read - 12 and 4...- the comma is misleading

Lines 232 -234 As I understand the simulation of the EA is done in a continuous dielectric medium (using the Polarizable Continuum Model (PCM)) and thus modifying the number of water molecules will limit the water bonding to the most favourable sites i.e. Ca rather than stabilising the cluster...perhaps this could be discussed in more detail.

Line 245 – the authors should not use such statements as considerably small – considerably smaller than what? – a range of radii of gyration need to be given – this would also then allow us to see what such sizes mean with respect to the critical nucleus size for the classical approach which are of the order of 2 nm [6]

Line 246 – 248 – I am not so sure trimers are never appear in the formation of C-S-H ...they are not present in the final structure but how do monomers or dimers become pentamers i.e. two dimers are bridged by a single Si monomer. Could a trimer form first and then a dimer add onto it? Wouldn't these results suggest that there is such a possibility and as an intermediate - and may even indicate step wise growth from complex or clusters to multi Si clusters, one Si by one Si unit (be it monomer, dimer or trimer)? In fact, in the work of Kumar et al [11] they mention that the amorphous globules, that the authors use as a key point in their arguments, does not follow the drierieketten rule – at Ca/Si ratio of around 1...again suggesting the trimers could be present in the amorphous globular phase.

From all of the similarity maps what I do not see is how these lowest energy clusters can be differentiated they all have some configurations close to zero relative energy, suggesting they are all possible and can "grow" and shrink into each other...i.e. a classical nucleation process...and therefore I would like to see some $\text{C}_3\text{S}_3\text{H}_x$ "clusters" made up from CHS_x and $\text{C}_2\text{S}_2\text{H}_x$ clusters could be interesting, a full study maybe beyond the scope of the current article but at least the trimer structures seen could be discussed in more detail.

Line 284- how did the authors choose the "best" clusters?

Line 298-300 -I could not find the formation enthalpies in my SI only those in figure 4 ?

Line 332 – indeed these structures of the $\text{C}_2\text{H}_2\text{H}_x$ and $\text{C}_4\text{S}_4\text{H}_2$ clusters resemble very well the C-S-H or tobermorite motif and is a key finding in the paper especially the $\text{C}_4\text{S}_4\text{H}_2$ unit.

Line 352 – 354 – so what is this size of the $\text{C}_2\text{S}_2\text{H}_{20}$ cluster ? This growth in such short times – towards what must be close to the critical nuclei size expected from CNT does not convince me that there is a stable cluster indicating non-classical nucleation pathway. As the authors say they just keep growing...the key step is the transformation or dehydration of the clusters that then give "solids" with the C-S-H like structures – after what I call the precipitation step, i.e. the formation of globules which

can probably be described by CNT as in [1].

Lines 370 – 385 – The proposal is reasonable that the C₂S₂H_x clusters dehydrate and become more ordered C-S-H units in time. Again, this gradual increase in particle size containing the C₂S₂H_x clusters suggests to me it follows a classical embryo growth process...which can be described by CNT as demonstrated for the nucleation of amorphous calcium carbonate (ACC) [1, 8].

To summarize

Three major point to consider

- The globules do not follow the Drierketten rule – so they can form via the energetically favourable trimers seen in these simulations.
- The authors base their mechanism a little too closely on ref 7 – where there are many assumptions and as far as I can see no precise control of pH which in an earlier paper by the group of ref 8 of the manuscript, pH has been seen as a key factor [16] as well as by Kumar et al [11] where the globules are stable for up to 24 hrs at low pH's (>11.5), so they are very dependent on the solution composition.
- The Ca-Si dimers do look like they could well be the building blocks...and the move to the tetramers – key in starting the C-S-H sheet structure is a very very important event. I think elongated C₄S₄H_x structures could then aggregate sheet to sheet to form the interlayers, maybe with the help of some of the monomers that are also seen in some of the low energy cluster situations...there is a high possibility that these insights could lead to demonstrating that C-S-H does in fact follow a non-classical nucleation pathway but the authors must be much much more clear about the pathway they suggest and links to the literature where this is clearly demonstrated rather than just making very general citations...i.e. it would be important to show a PNC that really is stable before transformation by dehydration.

References

- [1] A. Carino et al , *Cryst. Growth Des.*, 17 (4), pp 2006–2015, (2017), DOI: 10.1021/acs.cgd.7b00006
- [2] De Yoreo, J.J., et al., *Science*, 2015. 349(6247).
- [3] J. J. Thomas, et al, *Cem. Concr. Res.*, 2011, 41, 1257–1278.
- [4] G. W. Scherer and F. Bellmann, *Cem. Concr. Res.*, 2017, DOI:10.1016/j.cemconres.2016.07.017.
- [5] G. W. Scherer and F. Bellmann, *Cem. Concr. Res.*, 2017, DOI: 10.1016/j.cemconres.2016.07.017.
- [6] M.R.Andalibi, et al, *J.Mater.Chem.A*,6, 363-373, 2018. [https://DOI: 10.1039/c7ta08784e](https://doi.org/10.1039/c7ta08784e)
- [7] A. Carino et al, *Acta Biomaterialia* 74, 478-488, 2018
- [8] Henzler et al., *Sci. Adv.* 2018;4 (1) – eaa06283 : DOI: 10.1126/sciadv.aao6283
- [9] A. E. Nielsen, *Kinetics of Precipitation*, Pergamon Press Oxford, 1964.
- [10] A. Testino, et al *Chem. Mater.*, 2005, 17, 5346–5356.
- [11] A. Kumar, et al *J. Phys. Chem. C*, 2017, 121, 17188–17196.
- [12] S. Galmarini et al, *J. Phys. Chem. C* 120, 22407-22413 (2016)
- [13] I. Richardson – *Acat Cryst.*(2014) B70 903-923
- [14] Duque-Redondo, E., et al, *Cem. Concr. Res.* 156, 106784 (2022).
- [15] Kulik, D. A.; et al. *Comput. Geosci.*2013, 17, 1–24.
- [16] V. Kanchanason, J. Plank, *Cem. Concr. Res.* 102 (2017) 90–98. <https://doi.org/10.1016/J.CEMCONRES.2017.09.002>.

Response to the reviewers:

C-S-H nucleation pathway from atomistic simulations

Xabier M. Aretxabaleta, Jon Lopez-Zorrilla, Iñigo Etxebarria, and Hegoi Manzano

We would like to thank the reviewers for their positive evaluation of our paper. Their comments are certainly relevant and taking them into account has improved the manuscript.

Below we reply point by point to all the comments. In addition, two versions of the manuscript have been made, one with the changes and one final “clean” version. Note that we have rewritten extensively the text to make a more accurate introduction about the multistep nucleation vs non-classical nucleation and adapt our nomenclature to the indications by the reviewers. Despite the changes, the core ideas and findings of the manuscript did not change. We have also included additional plots and info in the S.I. Finally, we have kept running the MD simulations during the time we have revised the manuscript, so now we show data from 15ns, instead of the original 6ns. The longer simulations do not modify the results and discussion, yet the aggregate sizes seem to be more stabilized and the conclusions are now stronger.

Response to the reviewers' comments

Reviewer #1 (Remarks to the Author):

In this manuscript, Aretxabaleta et al present a vast collection of numerical data from free energy calculations of C-S-H prenucleation clusters (PNCs), specifically for Ca/Si ratio 1 for compositions $n=1, 2, 4$ and x amount of water molecules. The energetically favoured structures are identified and compared to bulk structure of tobermorite and a nucleation path is proposed.

C-S-H has various structures that depend on the Ca/Si ratio and it is mostly amorphous. The crystalline structure of tobermorite is often found in hardened cement paste but it is not evident that PNCs have to resemble tobermorite. Some recent experiments of Bae et al, Materials 2016, 9(12), 976 have reported Ca containing nanoparticles as precursors of C-S-H. Why not investigating a Ca/Si ratio around 1.7 that is the most common found in cement paste? I suppose the results will be very different. How can we know from this work that the PNCs identified here are most relevant ones for cement paste?

There are two important aspects in this comment, the resemblance of the PNCs with tobermorite, and the Ca/Si ratio of the cluster.

Regarding the first one, we are fully aware of the differences between tobermorite and the C-S-H. It is true that the C-S-H has a variable composition and it is not crystalline as tobermorite. However, there is a general consensus in the community about their similarity. Tobermorite is seen as the crystalline, ordered, ideal, or even the thermodynamically stable counterpart of the disordered, metastable C-S-H. Experimental studies use tobermorite-like models to rationalize their observations^{1 2}, and atomic-scale models of the C-S-H for simulations are built by modification of the tobermorite crystal³.

Therefore, it seemed logical for us to compare the C₄S₄H₂ PNC with tobermorite, to make the discussion in the paper simple. Tobermorite is easier to display due to its crystallinity, and, as we discuss in the paper, the same C₄S₄H₂ structure was obtained after relaxation of the cluster isolated from tobermorite crystal in ref⁴. But we can also compare the C₄S₄H₂ PNC with disordered C-S-H atomic models, or even with jennite (deprecated as C-S-H model years ago), as this basic motif is also present in those cases. After the reviewer's comment, we realized that a more detailed explanation will strengthen the plausibility of the proposed PNCs as the C-S-H basic building block. We have added figures in the S.I. equivalent to fig 4b, but for jennite and for a C-S-H model with a C/S = 1.67, and we have explained in the text that the C₄S₄H₂ PNCs can be found not only on tobermorite but also on C-S-H models and jennite crystals and discuss it in the text.

¹ Cuesta, A., Zea-Garcia, J. D., Londono-Zuluaga, D., De la Torre, A. G., Santacruz, I., Vallcorba, O., ... & Aranda, M. A. (2018). Multiscale understanding of tricalcium silicate hydration reactions. *Scientific reports*, 8(1), 8544.

² Black, L., Garbev, K., Stemmermann, P., Hallam, K. R., & Allen, G. C. (2003). Characterisation of crystalline CSH phases by X-ray photoelectron spectroscopy. *Cement and concrete research*, 33(6), 899-911.

³ Duque-Redondo, E., Bonnaud, P. A., & Manzano, H. (2022). A comprehensive review of CSH empirical and computational models, their applications, and practical aspects. *Cement and Concrete Research*, 156, 106784.

⁴ Manzano, H., Ayuela, A., & Dolado, J. S. (2007). On the formation of cementitious C-S-H nanoparticles. *Journal of computer-aided materials design*, 14, 45-51.

Regarding the Ca/Si ratio, we know that the average value for C-S-H in cement is ~ 1.7 . It must be noticed that we worked with a ratio of 1 for the EA, yet the MD simulations are done for a Ca/Si ratio of 1.5 to be closer to the experimental value. In the EA search, we focused on the Ca/Si = 1 to build clusters systematically. For the smaller system, CSHx, there is experimental ^{5 6 7} and computational ⁸ evidence about the presence of Ca/Si = 1 complexes. The general belief is that the prevalent silicate species in the pore solution are CaSiO4H2 complexes (log K = 4.6), and the extra Ca will be in the form of a hydrated Ca+2 ion. For larger clusters, there is no experimental information, so we assume as a working hypothesis that our CSH10 complexes will aggregate and form dimeric species C2S2Hx. It is true that at this stage we cannot rule out that additional Ca+2 ions will participate in the aggregation, forming dimeric clusters with Ca/Si > 1. However, given the structure of the C2S2Hx, we think that the existence of Ca/Si = 1 dimers could actually be one of the conclusions of the paper. We added this explanation to the methodology section.

From a more practical point of view, it must be recognized that this is the first work dealing with C-S-H PNCs beyond the smaller CSHx complex, and the first one using this multiscale methodology. We cannot explore all the possible variables in the C-S-H formation. Our EA search included ~ 18000 energy minimization calculations, first with empirical potentials and then with DFT. Exploring different C/S ratios will certainly be interesting, but the research should be conducted one step at a time.

⁵ Hummel, Wolfgang, et al. "Nagra/PSI chemical thermodynamic data base 01/01." *Radiochimica Acta* 90.9-11 (2002): 805-813.

⁶ Lothenbach, Barbara, and Frank Winnefeld. "Thermodynamic modelling of the hydration of Portland cement." *Cement and Concrete Research* 36.2 (2006): 209-226.

⁷ Tanaka, M., & Takahashi, K. (2002). Characterization of silicate monomer with sodium, calcium and strontium but not with lithium and magnesium ions by fast atom bombardment mass spectrometry. *Journal of mass spectrometry*, 37(6), 623-630.

⁸ Galmarini, S., Kunhi Mohamed, A., & Bowen, P. (2016). Atomistic simulations of silicate species interaction with portlandite surfaces. *The Journal of Physical Chemistry C*, 120(39), 22407-22413.

The data are presented via sketch-maps which is an elegant way to visualise such a large amount of data. In the SI, input files for the simulations are provided. Given the plethora of computational codes used and the type of data provided in this article, it is impossible to judge the soundness of the simulations, except if one repeats the simulations. The authors could have included a more typical simulation plot in the SI.

Actually, we believe that the collection of computational methodologies is one of the strengths of our research. We set the objective and then designed the research without any code/method limitation, and we just chose the most appropriate tools at each step. In principle, using more or fewer codes/methods does not compromise the soundness of the simulations, and the necessity of repeating the simulations to really assess their quality is an issue no matter how many codes/methods are used. As a matter of fact, we could say that it is the same scenario even for experimental works. To address it, we included in the S.I. input files for all the codes used in the paper, so the quality and correctness of our simulations can be checked. We also provided .json files ready to be loaded into chemscope, where the reader can have access to all the relaxed structures and the analyzed properties (energies, bond distances, coordinations, etc.) This is more transparent and reproducible than most of the simulation works published nowadays. Finally, after the first review round, the quality of the EA simulations is validated by Prof. Oganov (reviewer 2), who developed the methodology and the USPEX code that we use.

We think that the soundness of the simulations is proved with all the previous, but we have tried our best to include additional plots in the SI:

- Supplementary figure S1: Plots of the Ca and Si speciation as a function of the pH.
- Supplementary figures S2 to S10: Plots of the Energy vs the population for the EA structures of all sizes.
- Supplementary figure S18: Image of the clusters used in the MD simulation, the initial random configuration of one of the simulations, and the final configuration after the aggregation (15ns).
- Supplementary figure S19(a): Weighted Partial pair distribution functions $WabG(r)$. This analysis has been used to understand the contribution of each atomic pair to the total $G(r)$ and identify the peaks.
- Supplementary figure S19(b-d): Same data as figure 4 in the main manuscript, but showing the independent results from the 4 simulations and not only the average.

This article has noteworthy results for a specialized audience. The lack of comparison with experimental data and the specific choice of Ca/Si=1 make it difficult to convince that these PNCs will be relevant to the complex dissolution and nucleation processed in cement. Therefore, I wouldn't recommend publication to a broad audience journal.

With all respect, we disagree with the reviewer at this point. We agree that a specialized audience will be the main target (as for any study). However, our work is also a novel

contribution to nucleation of minerals/materials. For instance, looking to the comments of reviewers 3 and 4: the main controversial points are about the classical - non-classical nucleation statements, the definition of PNCs, etc. As discussed by reviewer 4, the difference between classical and non-classical is not straightforward at all, and reviewer 3 points out that there is no non-classical nucleation “theory”, just “models”. We are working at the forefront of a complex problem, and the study will have applications in cement, but it will also contribute to the general understanding of nucleation and will benefit other systems like ceramics, minerals, salts, silicates, nanoparticles...

Regarding the lack of comparison with experiments, it must be taken into account that the simulations focus on the structure and aggregation of C-S-H primary particles. These particles have not been experimentally characterized or detected, as their experimental determination is extremely challenging. The comparison is not possible at this point, but we try to put our results in context, and compare them with other systems (carbonates, sulfates...). In this revision, we have also added a PDF analysis of the aggregates and compared it with that of tobermorite and a bulk C-S-H model.

Finally, about the relevance of these PNCs to the complex dissolution and nucleation processed in cement, we do not claim to solve the puzzle and understand C-S-H nucleation. This work is a step forward, a quite novel and important one we think, which contributes significantly to the advance of our understanding. Clearly, more work will be necessary, and the final answer will be given by experiments and not by simulations. But we are quite far from that, and we are convinced that this work will pave the way for future research.

Some minor comments:

- in the figures nomenclature is used that hasn't been defined in the text or caption.
- few references are not in the correct format, ref. 3 and 4 are the same.

Thank you very much for spotting these mistakes and typos. We modified the manuscript to correct these points, and we revised again the whole manuscript and references.

Reviewer #2 (Remarks to the Author):

This is an impressive work employing a whole hierarchy of simulation methodologies in a clever way, in order to address the puzzles of the formation of cement. The work is important and appears to be correct. I recommend to publish it, with optional minor revision.

Thank you very much to reviewer 2 for his kind comments. We are particularly delighted about having such a positive review by Prof. Oganov, who developed the USPEX methodology and code that we are using in this work. It is certainly an advanced simulation technique, so we are proud to see that we are making good use of it.

Some comments:

1. I recommend to pay more attention to describing clearly the implications of the results. Where can we expect complex nucleation mechanisms? Can we use present results to improve cement?

It is always a hard task to describe the implications of results from atomistic simulations without overstatements. Especially in cement and concrete hydration, nucleation will be a process with multiple facets, and it is difficult to capture them with simulations. Even experiments on C-S-H nucleation are mainly done for synthetic samples and homogeneous nucleation, while in reality the C-S-H nucleation will entail the equilibrium of the pore solution between the dissolving and forming solids, will be mainly heterogeneous, and will happen in competition with the nucleation of other phases.

Nevertheless, we agree that the implications can and must be discussed. In the reorganization of the manuscript we have changed the "conclusions" section by a "discussion" section, more aligned with the format of the journal. There we do a summary of the proposed mechanism with a scheme as requested by reviewer 3, and the whole section tackles the implications of the work.

2. I spotted a typo on the bottom of p. 4 - should be H_3SiO_4^- , not H_3SiO_8^- .

Thank you very much for spotting this typo, we correct it.

3. The "sketch maps" were first developed in this paper, which would be correct to mention: Oganov A.R., Valle M. (2009). How to quantify energy landscapes of solids. J. Chem. Phys. 130, 104504.

Thank you very much for pointing out this reference. It is true that we missed the first utilization of a sketch map to classify structures. We have included the reference in the manuscript.

None of these minor comments decrease the quality of this work, which, as I wrote, should be accepted after minor revision.

Artem R. Oganov

Reviewer #3 (Remarks to the Author):

This study examines the initial stages of the formation of calcium silicate hydrate (CSH), the primary binding component of cement. The authors employed various atomic simulation techniques to better understand the clustering that occurs during the prenucleation stages of CSH formation. The study offers new perspectives on this topic, which align well with previously reported experimental and computational findings. However, it would be useful for the authors to provide a comprehensive illustration that integrates their results with those already reported in previous literature.

The manuscript is well-written and, given the significance of cement, will likely attract a wide range of readers. However, before it is published, there are a few aspects of the manuscript that could be enhanced, as outlined in the comments below.

Thank you very much to reviewer 3 for his positive evaluation and constructive feedback. He raised many interesting points that we have adopted in the manuscript. We will answer his detailed comments point by point.

Detailed comments:

Abstract.

L7, Recent evidences indicate that the C-S-H nucleation is non-classical, yet detailed information on the nucleation steps is still missing.

>> be more specific about which 'detailed information' is still missing. (both in the abstract and introduction)

Following the indications of reviewer 4 we have changed the "non-classical" term to "multistep". We cannot evaluate at this point if the nucleation is classical or non-classical as we did not evaluate the free energy evolution.

Nevertheless, the question still holds. When we wrote "detailed information" we meant "microscopic details" about the process. The characterization of the nucleation process in any material is very challenging due to the small size and lifetime of the transient species. So the main details that we don't know are (1) which are the dominant species and their characteristics, (2) their stability with respect to the species in solution and with respect to the final solid, and (3) how they merge/aggregate to form larger structures. We have specified these 3 points in the manuscript.

ORIGINAL: Recent evidences indicate that the C-S-H nucleation is non-classical, yet detailed information on the nucleation steps is still missing.

NEW: Recent evidence indicates that the C-S-H nucleation involves at least two steps, yet the underlying atomic scale mechanism, the nature of the primary particles and their stability, or how they merge/aggregate to form larger structures is unknown.

Introduction

L30. In non-classical nucleation (NCN), the exact number of steps and nucleation paths are system dependent, yet there are generalities. First, the existence of thermodynamically stable pre-nucleation clusters (PNCs).

>> Rather than a generality, I would say PNCs are a common feature in nonclassical nucleation pathways, but definitely not a necessity .

We agree with the reviewer on this point, his suggestion is more accurate than our original statement. Nonclassical nucleation strictly refers to 2 or more steps with their respective energy barriers, and those steps usually include PNCs, but not necessarily. Therefore we have changed the text:

ORIGINAL: In non-classical nucleation (NCN), the exact number of steps and nucleation paths are system dependent, yet there are generalities. First, the existence of thermodynamically stable pre-nucleation clusters (PNCs).

NEW: In contrast with the classical nucleation theory (CNT), multistep nucleation paths involve several (meta)stable states preceding the formation of stable nuclei. The exact number of steps and the nucleation paths are system dependent, yet there are some common features like the existence of thermodynamically stable pre-nucleation clusters (PNCs) or more generally metastable primary particles (PPs).

>> The authors should provide a definition of "PNCs", since formally these are clusters which form in solution conditions undersaturated with respect to the final phase (e.g. Gebauer et al, Science, 2008). In this respect it is not clear if the clusters discussed in this studies are PNC's sensu stricto (i.e. forming already at concentrations below the solubility of the final phase) or not?

The reviewer raised a good point. We don't know if the clusters in this work are formed below solubility conditions. From thermodynamic databases we have that CaSiO₄H₂ complexes have a solubility constant of log K = 4.6, and the C-S-H a log K > 10 (the exact value depends on the composition and the model adopted). Therefore, it is expected that at

least the complexes will form in undersaturated conditions. For larger clusters, we do not have data. Therefore, we cannot call these clusters PNCs in the most strict sense. To be more precise in our statements, we have changed the nomenclature from PNC to “primary particles (PPs)” in the text.

L31. The PNCs are small, formally do not have a phase interface, and should be considered solutes rather than solids.

>> define “small”

We took the definition of PNCs proposed in reference ⁹, as it illustrates well the structural features of PNCs in contrast to larger nanoclusters or nanoparticles. It is true that the “small” adjective is not precise, and the remaining part of the sentence is more important “formally do not have a phase interface, and should be considered solutes rather than solids”. Nevertheless, to define a size, we could say according to ref ¹⁰, that PNCs sizes could range between diameters of 0.5 and 2nm for calcium carbonate, and ref ¹¹ quantifies a particle size <3nm for calcium sulfate.

We also use now primary particles (PP) rather than PNCs because of the previous question

ORIGINAL: The PNCs are small, formally do not have a phase interface, and should be considered solutes rather than solids.

NEW: The PPs should be considered solutes rather than solids: they formally do not have an interface and their size ranges between 0.5 to 3nm.

L40. Direct experimental observation of PNCs structure and dynamics is extremely challenging due to the time and size scales. Atomistic simulations have identified the formation of dynamically ordered liquid-like oxyanion polymers (DOLLOPs) formed by ion pairs in Ca-phosphate and Ca-carbonate. However, for barite (BaSO₄) compact clusters resembling crystalline motifs have been reported.

>> Experimental observations of PNCs have been reported for calcium sulfate, where the formation and evolution of PNCs to the final phase has been monitored in situ (Stawski et al., Formation of calcium sulfate through the aggregation of sub-3 nanometre primary species. *Nat. Commun.*, 2016, 7, 11177) and a tentative structure of the PNCs, based on

⁹ Gebauer, D., Kellermeier, M., Gale, J. D., Bergström, L. & Cölfen, H. Pre-nucleation clusters as solute precursors in crystallisation. *Chem. Soc. Rev.* 43, 2348–2371, DOI: 10.1039/C3CS60451A (2014).

¹⁰ Pouget, E. M., Bomans, P. H., Goos, J. A., Frederik, P. M., de With, G., & Sommerdijk, N. A. (2009). The initial stages of template-controlled CaCO₃ formation revealed by cryo-TEM. *Science*, 323(5920), 1455-1458.

¹¹ Stawski, T. M., Van Driessche, A. E., Ossorio, M., Diego Rodriguez-Blanco, J., Besselink, R., & Benning, L. G. (2016). Formation of calcium sulfate through the aggregation of sub-3 nanometre primary species. *Nature Communications*, 7(1), 11177.

scattering data and MD, has been put forward (T. M. Stawski et al., The Structure of CaSO₄ Nanorods: The Precursor of Gypsum. *J. Phys. Chem. C* 123, 23151, 2019). I would be meaningful to include these results into this part of the introduction.

These papers are certainly relevant to our work. When we discuss the PNC structure we change the text as follows:

ORIGINAL: Direct experimental observation of PNCs structure and dynamics is extremely challenging due to the time and size scales. Atomistic simulations have identified the formation of dynamically ordered liquid-like oxyanion polymers (DOLLOPs) formed by ion pairs in Ca-phosphate and Ca-carbonate. However, for barite (BaSO₄) compact clusters resembling crystalline motifs have been reported.

NEW: Direct experimental observation of PNCs and PPs structure and dynamics is extremely challenging due to the time and size scales. For calcium sulfate, the formation and evolution of PNCs up to the final CaSO₄ x xH₂O polymorphs has been monitored in situ by X-ray small- and wide-angle scattering ¹², and rod-like particles have been reconstructed based on the experiments and MD simulations¹³. For Ca-phosphate and Ca-carbonate, atomistic simulations have also identified the formation of dynamically ordered liquid-like oxyanion polymers (DOLLOPs) formed by ion pairs in solution. In the mentioned cases, the PNCs and DOLLOPs do not have the exact structure of the final crystal phases, and it is suggested that a posterior ordering of larger aggregates is what defines the different crystalline polymorphs. However, for barite (BaSO₄) compact clusters resembling crystalline motifs have been reported after MD simulations in aqueous environments.

L47. In the particular case of synthetic C-S-H, dynamics light scattering (DLS), small-angle X-ray scattering (SAXS), and cryo-TEM have shown the formation of "amorphous solid spheroids" of about 60 nm that reorganize into C-S-H platelets after around 5h.

>> It would probably make sense to specify the conditions of these precipitation experiments (e.g. diluted solution, direct mixing of solutions,..).

¹² Stawski, T. M., Van Driessche, A. E., Ossorio, M., Diego Rodriguez-Blanco, J., Besselink, R., & Benning, L. G. (2016). Formation of calcium sulfate through the aggregation of sub-3 nanometre primary species. *Nature Communications*, 7(1), 11177.

¹³ Stawski, T. M., Van Driessche, A. E., Besselink, R., Byrne, E. H., Raiteri, P., Gale, J. D., & Benning, L. G. (2019). The structure of CaSO₄ nanorods: The precursor of gypsum. *The Journal of Physical Chemistry C*, 123(37), 23151-23158.

Instead of focusing on a single work ¹⁴, we added additional references that also identify the amorphous spheroids as a precursor of the C-S-H ^{15 16 17}. In the four studies, the C-S-H was synthetic, using different methods to mix a sodium silicate solution with a Ca nitrate solution under controlled conditions. As we will cite the 3 papers we are not going to enter into details of the individual methods, just mention that are synthetic from mixing solutions and not hydration of materials

ORIGINAL: In the particular case of synthetic C-S-H, dynamics light scattering (DLS), small-angle X-ray scattering (SAXS), and cryo-TEM have shown the formation of "amorphous solid spheroids" of about 60 nm that reorganize into C-S-H platelets after around 5h.

NEW: In the particular case of C-S-H synthesized from sodium silicate and calcium nitrate solutions ^{18 19}, dynamics light scattering (DLS), small-angle X-ray scattering (SAXS), and transmission electron microscopy (TEM) have shown two well differentiated steps: the formation of "amorphous solid spheroids" of about 50-60 nm that reorganize into C-S-H platelets with time.

Results

Therefore, the Ca/Si ratio is 1 and kept constant for all the clusters.

>> Indicate that this ratio is different from the typical Ca/Si ratio of CSH in cement pastes and provide a brief state of why this ratio was selected (could be added to Mat&Met).

This is a common comment from reviewers 1, 3, 4. We replied to reviewer 1, and we have added to the manuscript the following explanation:

NEW: Although the Ca/Si ratio for C-S-H in cement is ~1.7, the Ca/Si ratio for the EA was chosen to be 1 and kept constant for all the clusters, which deserves an explanation. For

¹⁴ Krautwurst, N., Nicoleau, L., Dietzsch, M., Lieberwirth, I., Labbez, C., Fernandez-Martinez, A., ... & Tremel, W. (2018). Two-step nucleation process of calcium silicate hydrate, the nanobrick of cement. *Chemistry of Materials*, 30(9), 2895-2904.

¹⁵ Schönlein, M., & Plank, J. (2018). A TEM study on the very early crystallization of CSH in the presence of polycarboxylate superplasticizers: Transformation from initial CSH globules to nanofoils. *Cement and Concrete Research*, 106, 33-39.

¹⁶ Kumar, A., Walder, B. J., Kunhi Mohamed, A., Hofstetter, A., Srinivasan, B., Rossini, A. J., ... & Bowen, P. (2017). The atomic-level structure of cementitious calcium silicate hydrate. *The Journal of Physical Chemistry C*, 121(32), 17188-17196.

¹⁷ Shen, X., Feng, P., Liu, X., Wang, W., Zhang, Y., Zhou, Y., & Ran, Q. (2023). New insights into the non-classical nucleation of CSH. *Cement and Concrete Research*, 168, 107135.

¹⁸ Schönlein, M., & Plank, J. (2018). A TEM study on the very early crystallization of CSH in the presence of polycarboxylate superplasticizers: Transformation from initial CSH globules to nanofoils. *Cement and Concrete Research*, 106, 33-39.

¹⁹ Kumar, A., Walder, B. J., Kunhi Mohamed, A., Hofstetter, A., Srinivasan, B., Rossini, A. J., ... & Bowen, P. (2017). The atomic-level structure of cementitious calcium silicate hydrate. *The Journal of Physical Chemistry C*, 121(32), 17188-17196.

the smaller clusters, CSH_x, there is experimental ^{20 21 22} and computational ²³ evidence about the existence of Ca/Si = 1 complexes. The general belief is that the prevalent silicate species in the pore solution are CaSiO₄H₂ complexes (log K =4.6), and the extra Ca will be in the form of a hydrated Ca⁺² ion. For larger clusters, there is no experimental information, so we assume as a working hypothesis that the CSH_x complexes will aggregate and form dimeric species C₂S₂H_x. We cannot rule out that additional Ca⁺² ions will participate in the aggregation, forming dimeric clusters with Ca/Si > 1. However, given the structure of the C₂S₂H_x, we think that the existence of Ca/Si = 1 dimers is plausible. Then, additional Ca will take part in the C-S-H structure in the aggregation stage. Consequently, for the MD simulations in section 3.5 we added Ca(OH)₂ in the solution to increase the Ca/Si ratio to 1.5, a value closer to the actual C-S-H composition.

>> Perhaps the authors could speculate on how a ratio higher than 1 would influence the prenucleation clusters structure?

It is hard to speculate. We plan to perform a similar EA search for other Ca/Si ratios in the near future and compare the energies of formation. Nevertheless, our first hypothesis as we stated in the previous question is that the Ca/Si = 1 dimers are the basic building block formed from the aggregation of the monomeric complexes, and additional Ca⁺² would be separately in solution. C-S-H with Ca/Si > 1 would form from the aggregation of dimers together with Ca in solution, but the stable PPs would be dimers with Ca/Si = 1.

We do not have data to support this speculation, so we prefer not to write it down in the manuscript.

Discussion

L286: in the context of non-classical nucleation theory.

>> At present there is not yet a formal theory for non-classical nucleation. More correct would be to say: non-classical nucleation models. (as far as I know the only theoretical framework dealing with non-nucleation can be found in the works of Lutsko: e.g. Science advances 5 (4), eaav7399; New Journal of Physics 20 (8), 083019)

We agree with the reviewer. We used the term “non-classical nucleation theory” just for direct comparison with the common CNT. But it is true that there is no formal theory and

²⁰ Hummel, Wolfgang, et al. "Nagra/PSI chemical thermodynamic data base 01/01." *Radiochimica Acta* 90.9-11 (2002): 805-813.

²¹ Lothenbach, Barbara, and Frank Winnefeld. "Thermodynamic modelling of the hydration of Portland cement." *Cement and Concrete Research* 36.2 (2006): 209-226.

²² Tanaka, M., & Takahashi, K. (2002). Characterization of silicate monomer with sodium, calcium and strontium but not with lithium and magnesium ions by fast atom bombardment mass spectrometry. *Journal of mass spectrometry*, 37(6), 623-630.

²³ Galmarini, S., Kunhi Mohamed, A., & Bowen, P. (2016). Atomistic simulations of silicate species interaction with portlandite surfaces. *The Journal of Physical Chemistry C*, 120(39), 22407-22413.

therefore we adopt the nomenclature suggested by the reviewer. We have changed “non-classical nucleation theory” to “non-classical nucleation models”.

L333: two typos ...in the CSH.... five of MD....

Thank you very much for spotting these typos, we corrected them.

L356: The aggregates are strongly solvated by water molecules, so they tend to be elongated. Figure 4(e) shows the asphericity of the clusters as a function of the gyration radius. Clearly, larger clusters tend to be more elongated, resembling the DOLLOPs found during CaCO₃ prenucleation.

>> Any reason why aggregates tend to elongate due to solvation?

>> SAXS measurements have revealed that in the case of CaSO₄ clusters are also elongated (Stawski et al., Nat. Commun., 2016, 7, 11177).

Elongated shapes maximize the surface to bulk ratio, and in the case of PPs elongated shapes will maximize the possible H₂O-Ca coordination. However, there are points to clarify in our statement:

First, we will remove the “strongly” adjective for the sentence. The term “strong” usually refers to the bonding strength, and it is well known that the Ca-H₂O bond is rather weak compared to other common metals. That can be quantified from the exchange rate of water molecules between the first hydration shell of the metals and the solution, which is faster for Ca than for other divalent cations like Zn, Mn or Mg²⁴. Therefore, the solvation is not exactly strong. We meant that the number of water molecules directly coordinated to Ca cations from the cluster is large.

Second, why should the cluster maximize the solvation with water? We think that the solvation must play an important role in stabilizing the small PPs and aggregates fulfilling the coordination spheres of Ca atoms. Then, the PPs will aggregate into larger structures and lose solvation water to form solid particles. This is actually very similar to what has been proposed for Ca-sulphate²⁵. If such a nucleation mechanism is correct, it makes sense that solvation plays a role in the stabilization of the PPs and aggregates, and therefore these will tend to be elongated to maximize the interactions.

²⁴ Ohlin, C. A., Villa, E. M., Rustad, J. R., & Casey, W. H. (2010). Dissolution of insulating oxide materials at the molecular scale. *Nature materials*, 9(1), 11-19.

²⁵ Stawski, T. M., Van Driessche, A. E., Ossorio, M., Diego Rodriguez-Blanco, J., Besselink, R., & Benning, L. G. (2016). Formation of calcium sulfate through the aggregation of sub-3 nanometre primary species. *Nature Communications*, 7(1), 11177.

We have modified the text to include the above points, as well as the suggested reference about CaSO₄ which proposes elongated rod-like clusters as primary particles.

ORIGINAL: The aggregates are strongly solvated by water molecules, so they tend to be elongated. Figure 4(e) shows the asphericity of the clusters as a function of the gyration radius. Clearly, larger clusters tend to be more elongated, resembling the DOLLOPs found during CaCO₃ prenucleation.

NEW: Figure 4(e) shows the asphericity of the clusters as a function of the gyration radius. Clearly, larger clusters tend to be more elongated, in agreement with the structure of the DOLLOPs and rod-like particles proposed as primary particles for CaCO₃ and CaSO₄ nucleation. The elongated shapes maximize the Ca-water coordination, which might stabilize these aggregates until they assemble into larger morphologies. A similar mechanism has been proposed for the nucleation of gypsum²⁶, with primary species that form aggregates without any coherent diffraction signal despite their large size, which suggests a disorder arrangement. The solvation water that we observe in this work may be the reason for the misalignment and disorder between the aggregates.

L381: Despite only observing the initial steps of hydration, we suggest that C₄S₄H₂ clusters will be progressively formed within the aggregates leading to crystallization, following the experimental steps found experimentally by Krautwurst et al. 7

>> it would be insightful if the authors could include a schematic figure of the nucleation pathway of CSH, combining their results with those previously reported. In addition, this could serve to detail what information is still missing to establish a holistic view of CSH nucleation.

We have done our best to make a schematic figure illustrating how the proposed nucleation pathway fits within the multistep nucleation process reported for synthetic C-S-H. Based on the turbidity vs time scheme to illustrate the stages²⁷, we have included in the figure where the proposed PPs fit within the nucleation, and the aggregation/dehydration stages to form

²⁶ Stawski, T. M., Van Driessche, A. E., Ossorio, M., Diego Rodriguez-Blanco, J., Besselink, R., & Benning, L. G. (2016). Formation of calcium sulfate through the aggregation of sub-3 nanometre primary species. *Nature Communications*, 7(1), 11177.

²⁷ Krautwurst, N., Nicoleau, L., Dietzsch, M., Lieberwirth, I., Labbez, C., Fernandez-Martinez, A., ... & Tremel, W. (2018). Two-step nucleation process of calcium silicate hydrate, the nanobrick of cement. *Chemistry of Materials*, 30(9), 2895-2904.

amorphous C-S-H precursor and the final foil-like structures (images from ²⁸) and the silicate chain information from ²⁹.

Alexander.

Reviewer #4 (Remarks to the Author):

General comments

A very interesting paper on a key topic for better understanding the earliest stages of the formation of Calcium Silicate Hydrate (C-S-H), the most abundant man-made material on earth. Despite many years of research there are still many unknowns with respect to C-S-H

²⁸ Schönlein, M., & Plank, J. (2018). A TEM study on the very early crystallization of CSH in the presence of polycarboxylate superplasticizers: Transformation from initial CSH globules to nanofoils. *Cement and Concrete Research*, 106, 33-39.

²⁹ Shen, X., Feng, P., Liu, X., Wang, W., Zhang, Y., Zhou, Y., & Ran, Q. (2023). New insights into the non-classical nucleation of CSH. *Cement and Concrete Research*, 168, 107135.

and this paper definitely contributes an interesting and original approach to the earliest moment of formation – pre-nucleation species from atomistic scale modeling.

I think the main limitation of the paper as submitted is the introduction. The authors do not set the scene adequately for what follows. My opinion is that the non-classical nucleation pathway suggested in reference 7 is a very circumspect proposal and is by no way proven or currently accepted in the community – it is just a possible interpretation of their data. Other interpretations could include a classical nucleation pathway with the first product being amorphous – as for amorphous calcium carbonate[1]. I think the authors should say that it is not clear from 7 that it is non-classical – albeit a possibility and then use their simulations to show in a first step how atomistic simulations can help contribute to unraveling the precipitation/nucleation pathway of C-S-H.

The paper is novel and I think breaks barriers that have not been broken before and promises to lead us towards a much better and atomistic scale view of C-S-H nucleation. However as stated above and discussed in the detailed comments below I feel the focus of the paper needs adjusting and many minor points need clarification. Therefore, my decision would be to accept but with major revisions.

Thank you very much for the positive comments. There is still a lot of work to do if we want to understand (and control) C-S-H nucleation and cement hydration, but we also think that this work brings a novel approach and opens new paths for research in the interpretation of experimental data.

Detailed comments

Line 6 I think C-S-H should be defined here as this is not a specialist cement journal – Calcium Silicate Hydrate

A definition of the C-S-H nomenclature has been added to the text:

OLD: One of the most crucial ones is the nucleation and growth of the C-S-H, the main component of the hardened cement paste.

NEW: One of the most crucial ones is the nucleation and growth of the C-S-H, the main component of the hardened cement paste. C-S-H is the acronym for hydrated calcium silicate, a XRay amorphous phase of variable composition $(\text{CaO})_x(\text{SiO}_2)_y(\text{H}_2\text{O})_z$ with a layered structure that resembles the tobermorite family of minerals.

Line 6 abstract – evidences – should read – evidence

Thank you for spotting the typo, we corrected it

Line 6 abstract – I am not convinced that the nucleation is non-classical. It might be multi-step but there is plenty of evidence for a nucleation energy barrier, albeit possibly towards an amorphous first phase but will come back that in more detail below.

We answer this challenging point together with the next one

Lines 26-30 – so here we go ...the Non-classical nucleation misunderstandings; are the authors (and the references cited) really talking about non-classical nucleation theory or just indirect routes...which still have an energy barrier...see [1] SI for a very clear discussion and [2] for a clear schematic of such pathways. i.e. unknown activation energies (classical pathways) are not non-classical. Many people induce the idea of a non-classical route when in fact what they mean is unknown formation pathway. Also, one has to be very clear about nucleation pathways and growth (atomistic, molecular, cluster, or aggregation (disordered and self-ordered)). Understood that both ref 7 and ref 8 in the manuscript show that an initial amorphous globular phase transforms into the more recognisable sheet like structure of synthetic C-S-H but there is no strong evidence in either paper that this amorphous first phase does not have a nucleation barrier. In fact, there are many papers indicating that there is such a classical nucleation barrier and go on to very convincingly analyze the primary and secondary nucleation rates according to classical nucleation theory (CNT) [3-6]. There are many examples of classical nucleation routes that give an amorphous and/or extra hydrated first product which then transforms into the more thermodynamically stable phase later in the process as the saturation of the solution is modified ([1,7, 8]). I think many people have jumped on this non-classical nucleation band wagon for growth processes that do not follow the atomic growth unit pathway i.e. the mesoparticles made up of nanoparticles via aggregation steps [2] and often actually do not look at the nucleation process, which is a non-trivial process to investigate. I strongly disagree with the review cited by the authors Lee et al ref 10 – that having a non-crystalline first product is a non-classical nucleation process... the seminal work by Nielsen does not stipulate the product has to be crystalline [9, 1]. That is to say the first product does not have to be crystalline and the transformation from amorphous to crystalline can be classical...i.e. a classical nucleation barrier for the first crystalline seed then growth takes over [1,2,8] or non-classical-via aggregation or dehydration steps [2]. So, for the moment I see no evidence in the citations 7 and 8 which indicate that C-S-H nucleation is non-classical – it is just an interpretation. Also, in the applied cement systems used to make concrete etc the dominant mechanism is probably heterogeneous nucleation. I understand that sub-critical size nuclei – embryos can adsorb and become stable nuclei and thus promote growth, akin to the secondary nucleation process suggested for several systems where nucleation nonetheless can be described in classical nucleation theory (CNT)[1,3,10]. I feel that the authors may in this paper be presenting the first real evidence of a non-classical nucleation pathway and thus they really need to get the introduction right.

Overall, we agree with the reviewer. At this stage, our simulations cannot determine the nature of the nucleation pathway, if it is classical or non-classical. We do not contribute to that discussion as we did not explore the free energy evolution. Our objective was to study the possible structure of the first C-S-H clusters, (whether they are ion pairs, complexes, primary particles, or PNCs) and their aggregation. The introduction aimed to point out the importance of these species in the nucleation, and for that, we adopted non-classical nucleation models in which those PNCs are a key ingredient. As the reviewer correctly indicates, we were thinking of multi-step or indirect nucleation routes rather than non-classical routes.

We read carefully the suggested references^{30 31}, and we realized that the identification of multi-step and non-classical as the same thing is an oversimplification made in many sources, but incorrect. We were falling into the same mistake. Therefore, we have rewritten the introduction to take all these points into account. Essentially, we mention that primary particles (PPs) have been identified as key species in the early stages of nucleation processes for many systems, including some related to C-S-H as zeolites, CaCO₃, CaSO₄, Ca₂PO₃, etc. We explain that they may be present both in classical or non-classical nucleation routes, and we highlight the importance of these species: the technological relevance as nucleation kinetics and polymorphism may be controlled by their (de)stabilization.

Regarding our results, the interpretation does not change, we just put them into the correct context. We agree that refs 7 and 8 do not strictly indicate non-classical nucleation but a multistep pathway.

Lines 61- 66 – I think the use of atomistic simulations for “pre-nucleation” species or clusters is a very important approach and will gain weight in coming years. However, I think that solution species – ion-pairs or hydrated ion pairs – are often in equilibrium with the simple ionic species that go to make them up. Such complexes in solution that can be the classical building block of the precipitate if this is the dominant(i.e. highest population) species in the solution.

As discussed above, we oversimplified the topic a bit. In the revised version we will try to be more careful with the use of PNCs. We have changed the nomenclature and we use here complex for the smallest size, primary particles (PPs) for the larger clusters, and we maintain aggregates for the assembly of PPs during the MD simulations.

³⁰ Carino, A., Testino, A., Andalibi, M. R., Pilger, F., Bowen, P., & Ludwig, C. (2017). Thermodynamic-kinetic precipitation modeling. A case study: The amorphous calcium carbonate (ACC) precipitation pathway unravelled. *Crystal Growth & Design*, 17(4), 2006-2015.

³¹ De Yoreo, J. J., Gilbert, P. U., Sommerdijk, N. A., Penn, R. L., Whitlam, S., Joester, D., ... & Dove, P. M. (2015). Crystallization by particle attachment in synthetic, biogenic, and geologic environments. *Science*, 349(6247), aaa6760.

Lines 69-72 – The two step route reported in reference 26 (the author’s own work) is really far from the non-classical route discussed here, so I am not too sure how relevant it is if the current work does not support their assumptions in ref 26. Despite the elegant calculations presented in ref 26 there is no evidence for this calcium hydroxide intermediate phase in any of the synthetic C-S-H synthesis papers that I am aware of, the references 7 and 8 certainly do not, the globules contain Ca/Si. In fact, the C-S-H globular particles were first clearly seen in the work of Kumar et al ([11] – Fig S3) and they show clearly that the globules are not amorphous silica or amorphous calcium hydroxide with high resolution (3nm spot size Table S2). So I am very skeptical about this calcium hydroxide 2 step mechanism.

Our work reported in ref ³² is indeed a very different nucleation route to the one proposed here. We studied that option because the role of portlandite (CH) in the C-S-H formation has been speculated for a long time. There is a clear thermodynamic relationship via the pore solution saturation, but the possibility of a transformation from CH to C-S-H has also been specifically suggested. As a matter of fact, the pozzolanic reaction between silica additions and CH has been reported as the process to form additional C-S-H, reducing porosity and increasing mechanical properties in the cement paste.

We agree that the route that we explore in this work is more likely to be closer to the actual one, at least in the synthetic conditions of the experiments that we refer to. However, the CH to CSH transformation may still be relevant in certain conditions: as a complementary mechanism when the Ca saturation is very high, in the cement paste as a topochemical transition in pozzolanic reactions, or as a partial transformation of CH when CSH heterogeneous nucleation takes place in its surface. Mentioning our previous work in the introduction could be relevant to the reader.

Line 111 is the last word in this line cluster ? it should read clusters..

Thank you very much, we have corrected it.

Line 141 – I am not too sure what the normalization to CS – CaO-SiO₂ really means wrt to the water that is then added...is this meant to be a stable anhydrous CaO -SiO₂ cluster or just the CaO and SiO₂ separate energies?

Comparing energies across different compositions and water contents is not trivial. We decided to present the energies in figures 1, 2, and 3 in kJ per mol of CaO-SiO₂, which is the “unit” that we repeat over. The water content is not taken into account, as the number of water molecules changes from size to size, and also the water directly coordinated to the

³² Aretxabaleta, X. M., López-Zorrilla, J., Labbez, C., Etxebarria, I., & Manzano, H. (2022). A potential CSH nucleation mechanism: atomistic simulations of the portlandite to CSH transformation. *Cement and Concrete Research*, 162, 106965.

Ca atoms may be different for the same size. In short, using the nomenclature C_nSnH_x , we present the energy in kJ/mol of CS, and therefore for $n = 2, 4$ we divide the total internal energy of the cluster by 2 and 4.

In any case, the energies across sizes are not comparable due to the variable water content. What makes sense is the free energy differences in figure 4, where we take fully into account the stoichiometry of the reactions in the calculations.

Lines 148 – 150 – it should be made clear that these are the species for separate calcium and silicate solutions...once mixed it is well accepted that the main silicate species becomes the neutral complex, $CaSiO_2(OH)_2$, at pH's above 13 with the $H_2SiO_4^{-2}$ being an order of magnitude less in concentration.[12]

True, it may be confusing. We have specified it in the text that we refer to calcium and silicate solutions separately. In addition, we have added to the S.I. custom plots of the silicate and Ca speciation as a function of pH based on references^{33,34,35}.

Line 164 – notice – should read – noted

Thank you very much, we have corrected it.

Lines 183 -186- indeed the 7-fold coordination of Ca is a key structural feature seen long before these recent SXRPRD studies [13] – indeed it is a key motif in the Tobermorite structure, the basis of most C-S-H models [13, 14].). Generally, coordination of less than 6 is rarely seen for calcium in the C-S-H system [13].

Yes, we agree with the reviewer, the 7-fold coordination is well known before the recent studies SXRPRD studies, and typically observe in the tobermorite family. We meant that it is used in SXRPRD studies to characterize the time development of the tobermorite-like structure. We have corrected the references, because the ref 25 and 26 in the original manuscript do not match the actual references we wanted to include, and we have included references to highlight that it is not new knowledge

ORIGINAL: It is important to notice that many complexes within accessible energies present 7-coordinated atoms (see figure 1c). That is the coordination of Ca atoms in the

³³ Eikenberg, J. On the problem of silica solubility at high ph. Tech. Rep., Paul Scherrer Inst.(PSI) (1990).

³⁴ Kutus, B. et al. A comprehensive study on the dominant formation of the dissolved $Ca(OH)_2(aq)$ in strongly alkaline solutions saturated by $Ca^{(ii)}$. RSC advances 6, 45231–45240 (2016).

³⁵ Šefcík, J. & McCormick, A. V. Thermochemistry of aqueous silicate solution precursors to ceramics. AIChE J. 43,2773–2784 (1997).15/15

C-S-H gel intralayer region, and it is common to use such a structural feature to characterize the early formation of C-S-H from Synchrotron X-ray powder diffraction (SXRPD). Our simulations indicate that the complexes may already display such coordination, and therefore they may interfere with the characterization of the solid phase.

NEW: It is important to notice that many complexes within accessible energies present 7-coordinated atoms (see figure 1c). That is the coordination of Ca atoms in the C-S-H gel intralayer region^{36,37}, and therefore is common to use such a structural feature to characterize the early formation of C-S-H for example in Synchrotron X-ray powder diffraction (SXRPD)^{38,39}. Our simulations indicate that the complexes may already display such coordination, and therefore they may interfere with the characterization of the solid phase.

Lines 190-197 the structures of the CSH complexes could also be compared to the much earlier work of Galmarini et al [12] where I believe $\text{CaSiO}_2(\text{OH})_2$ was first demonstrated to be a stable complex, in particular the Ca-Si distance compared to the results in the current manuscript. The complex is in equilibrium with the solution and its concentration (or activity) can be calculated under different solution conditions using relatively standard thermodynamic packages such as GEMS [15]. Although the paragraph is headed CSHx complexes ...the authors use the description cluster quite often in the paragraph and even in the Figure 1 caption...I think it is very important that this difference between expected complexes in thermodynamic equilibrium with the solution and the proposed clusters is maintained. That is to say, the clusters that may represent a stable thermodynamic state but are smaller than a perceived stable nucleus that would then grow and reduce its free energy when increasing its size according to the CNT.

We knew the paper by Galmarini et al.⁴⁰, but we always refer to that paper in relation to the silicate-CH interaction. We overlooked the first part regarding the complex, which is indeed very interesting and relevant to our work.

³⁶ Manzano, H., Ayuela, A., & Dolado, J. S. (2007). On the formation of cementitious C–S–H nanoparticles. *Journal of computer-aided materials design*, 14, 45-51.

³⁷ Aretxabaleta, X. M., López-Zorrilla, J., Labbez, C., Etxebarria, I., & Manzano, H. (2022). A potential CSH nucleation mechanism: atomistic simulations of the portlandite to CSH transformation. *Cement and Concrete Research*, 162, 106965.

³⁸ Grangeon, S., Fernandez-Martinez, A., Baronnet, A., Marty, N., Poulain, A., Elkaim, E., ... & Claret, F. (2017). Quantitative X-ray pair distribution function analysis of nanocrystalline calcium silicate hydrates: a contribution to the understanding of cement chemistry. *Journal of applied crystallography*, 50(1), 14-21.

³⁹ Cuesta, A., Santacruz, I., Angeles, G., Dapiaggi, M., Zea-Garcia, J. D., & Aranda, M. A. (2021). Local structure and Ca/Si ratio in CSH gels from hydration of blends of tricalcium silicate and silica fume. *Cement and Concrete Research*, 143, 106405.

⁴⁰ Galmarini, S., Kunhi Mohamed, A., & Bowen, P. (2016). Atomistic simulations of silicate species interaction with portlandite surfaces. *The Journal of Physical Chemistry C*, 120(39), 22407-22413.

In this version, we have compared the $\text{CaSiO}_2(\text{OH})_2$ complex structure from Galmarini et al. with the structures obtained from EA.

Line 228 – 12, 4 – should read - 12 and 4...- the comma is misleading

Thank you very much, we have corrected it.

Lines 232 -234 As I understand the simulation of the EA is done in a continuous dielectric medium (using the Polarizable Continuum Model (PCM)) and thus modifying the number of water molecules will limit the water bonding to the most favourable sites i.e. Ca rather than stabilising the cluster...perhaps this could be discussed in more detail.

The EA is done in vacuum because PCM models are not available in SIESTA. We tried to use Gaussian or Orca together with USPEX to include the effect of the implicit solvent directly into the EA search, but we found that both software had important problems to minimize high-energy structures. The simulations were not converged or even crashed in many cases, even after the ReaxFF pre-optimization, making them impractical for the search. We found SIESTA more robust and therefore we stick to it despite not having PCM options. As a note, our tests indicate that VASP or Quantum Espresso were as robust as SIESTA, but for clusters with a large vacuum space, the plane waves made them slower for the desired accuracy.

In our work, the solvent effect using PCM is included in the Gaussian simulations, together with the 0K vibrational analysis to get a grasp of the free energy.

Line 245 – the authors should not use such statements as considerably small – considerably smaller than what? – a range of radii of gyration need to be given – this would also then allow us to see what such sizes mean with respect to the critical nucleus size for the classical approach which are of the order of 2 nm [6]

True, that is a qualitative statement that can be confusing. We want to express that the gyration radius is smaller for the lowest energy PP than for high energy ones- We replaced the sentence:

ORIGINAL: Nevertheless, the lowest energy structures within region I share similar characteristics: they are formed by a silicate monomer and a silicate trimer, and their gyration radius is considerably small.

NEW: Nevertheless, the lowest energy structures within region I share similar characteristics: they are formed by a silicate monomer and a silicate trimer, and their gyration radius is < 0.3 nm, at the lower end of the range for all the structures.

Line 246 – 248 – I am not so sure trimers are never appear in the formation of C-S-H ...they are not present in the final structure but how do monomers or dimers become pentamers i.e. two dimers are bridged by a single Si monomer. Could a trimer form first and then a dimer add onto it? Wouldn't these results suggest that there is such a possibility and as an intermediate - and may even indicate step wise growth from complex or clusters to multi Si clusters, one Si by one Si unit (be it monomer, dimer or trimer)? In fact, in the work of Kumar et al [11] they mention that the amorphous globules, that the authors use as a key point in their arguments, does not follow the dreierketten rule – at Ca/Si ratio of around 1...again suggesting the trimers could be present in the amorphous globular phase.

From all of the similarity maps what I do not see is how these lowest energy clusters can be differentiated they all have some configurations close to zero relative energy, suggesting they are all possible and can “grow” and shrink into each other...i.e. a classical nucleation process...and therefore I would like to see some C₃S₃H_x “clusters” made up from CHS_x and C₂S₂H_x clusters could be interesting, a full study maybe beyond the scope of the current article but at least the trimer structures seen could be discussed in more detail.

If we understand reference 11, it was not possible to fit the NMR data assuming a dreierketten model for the globules at low Ca/Si ratios. Other evidence (including one published during the review time) also suggests that in the amorphous globules not even chains longer than dimers appear (characterized by Q₂ peaks in Si-NMR) but also branched silicate oligomers (characterized by Q₃ peaks in NMR)^{41 42} and silicate monomers (q₀ peaks). The silicate polymerization (Q₁, Q₂ and Q₃ peaks) was recorder after just 1 minute of reaction. Then, Q₀ and Q₃ peaks disappear when the globules are transformed into foil-like crystallites. While the methodology to stop the reaction at 1min, 1hour, etc might affect the aggregation of the globules (their size) and the polymerization of the silicate structures, the studies suggest that at least at low Ca/Si ratio in synthetic C-S-H, trimers, and other silicate oligomers may be present in the amorphous precursor.

Our results fit even better with these observations than with the existence of only dimers. Therefore, the possibility of trimers being present in the primary species cannot be discarded at all, and we will highlight it in the text and in the proposed nucleation path.

Line 284- how did the authors choose the “best” clusters?

⁴¹ Plank, J., Schönlein, M., & Kanchanason, V. (2018). Study on the early crystallization of calcium silicate hydrate (CSH) in the presence of polycarboxylate superplasticizers. *Journal of Organometallic Chemistry*, 869, 227-232.

⁴² Shen, X., Feng, P., Liu, X., Wang, W., Zhang, Y., Zhou, Y., & Ran, Q. (2023). New insights into the non-classical nucleation of CSH. *Cement and Concrete Research*, 168, 107135.

According to the EA, the best structures are those which have the best value of the fit function. In our case, the fit function is the enthalpy, so we assume that the best structures are the lowest enthalpy ones.

Line 298-300 -I could not find the formation enthalpies in my SI only those in figure 4 ?

We forgot to include the data in the SI. In the revised version we include it.

Line 332 – Indeed these structures of the C₂H₂H_x and C₄S₄H₂ clusters resemble very well the C-S-H or tobermorite motif and is a key finding in the paper especially the C₄S₄H₂ unit.

We are glad with this result. EA explored the configurational space without a priori information or constraints, and the fact that the lowest energy structures that we found encode the structural motifs of tobermorite and the C-S-H is significant.

Line 352 – 354 – so what is the size of the C₂S₂H₂₀ cluster? This growth in such short times – towards what must be close to the critical nuclei size expected from CNT does not convince me that there is a stable cluster indicating non-classical nucleation pathway. As the authors say they just keep growing...the key step is the transformation or dehydration of the clusters that then give “solids” with the C-S-H like structures – after what I call the precipitation step, i.e. the formation of globules which can probably be described by CNT as in [1].

We assume that the reviewer refers to the size of the aggregate formed by the assembly of C₂S₂H₂₀ clusters. To answer the question, the radius of gyration of the largest aggregate is ~ 1.5 nm.

Regarding the comments, we will reply to this in the following question.

Lines 370 – 385 – The proposal is reasonable that the C₂S₂H_x clusters dehydrate and become more ordered C-S-H units in time. Again, this gradual increase in particle size containing the C₂S₂H_x clusters suggests to me it follows a classical embryo growth process...which can be described by CNT as demonstrated for the nucleation of amorphous calcium carbonate (ACC) [1, 8].

As we have discussed through the review, it is true that we cannot demonstrate a non classical nucleation process because we do not evaluate free energy barriers. We knew it, but we used non-classical nucleation to actually refer to multistep, which might not be correct despite the recurrent use in the literature. So we agree with this and the previous comments, and we changed in the text non-classical nucleation by multistep nucleation.

Regarding the continuous growth of the aggregates, it must be taken into account that we started our MD simulations at the (estimated) concentration of silicates within the amorphous spheroids. If the experimental estimation is correct, then we are doing simulations at a concentration in which the C₂S₂H₂₀ clusters should not exist and should be already aggregated into an amorphous solid structure. Such a supersaturation implies that the aggregation in our simulation is fast, of the order of 10ns, but the actual kinetic process may be more gradual. It will be interesting to test in the future the free energy barrier of aggregation at different concentrations to evaluate if there are relevant energy barriers and explore saturation ranges.

To summarize

Three major point to consider

- The globules do not follow the Drierketten rule – so they can form via the energetically favourable trimers seen in these simulations.

As mentioned above, we have taken this into account from the perspective of the suggested reference ⁴³ and the recently published work ⁴⁴, adding a paragraph about the possibility of trimers as low energy PPs or transient species.

- The authors base their mechanism a little too closely on ref 7 – where there are many assumptions and as far as I can see no precise control of pH which in an earlier paper by the group of ref 8 of the manuscript, pH has been seen as a key factor [16] as well as by Kumar et al [11] where the globules are stable for up to 24 hrs at low pH's(>11.5), so they are very dependent on the solution composition.

We have modified the description of the nucleation process, from non-classical to multistep mechanism, and we have added references and discuss other works that report very similar structures ^{45 46 47}.

- The Ca-Si dimers do look like they could well be the building blocks...and the move to the tetramers – key in starting the C-S-H sheet structure is a very very important event. I think

⁴³ Kumar, A., Walder, B. J., Kunhi Mohamed, A., Hofstetter, A., Srinivasan, B., Rossini, A. J., ... & Bowen, P. (2017). The atomic-level structure of cementitious calcium silicate hydrate. *The Journal of Physical Chemistry C*, 121(32), 17188-17196.

⁴⁴ Shen, X., Feng, P., Liu, X., Wang, W., Zhang, Y., Zhou, Y., & Ran, Q. (2023). New insights into the non-classical nucleation of CSH. *Cement and Concrete Research*, 168, 107135.

⁴⁵ Kumar, A., Walder, B. J., Kunhi Mohamed, A., Hofstetter, A., Srinivasan, B., Rossini, A. J., ... & Bowen, P. (2017). The atomic-level structure of cementitious calcium silicate hydrate. *The Journal of Physical Chemistry C*, 121(32), 17188-17196.

⁴⁶ Shen, X., Feng, P., Liu, X., Wang, W., Zhang, Y., Zhou, Y., & Ran, Q. (2023). New insights into the non-classical nucleation of CSH. *Cement and Concrete Research*, 168, 107135.

⁴⁷ Schönlein, M., & Plank, J. (2018). A TEM study on the very early crystallization of CSH in the presence of polycarboxylate superplasticizers: Transformation from initial CSH globules to nanofoils. *Cement and Concrete Research*, 106, 33-39.

elongated C₄S₄H_x structures could then aggregate sheet to sheet to form the interlayers, maybe with the help of some of the monomers that are also seen in some of the low energy cluster situations...there is a high possibility that these insights could lead to demonstrating that C-S-H does in fact follow a non-classical nucleation pathway but the authors must be much much more clear about the pathway they suggest and links to the literature where this is clearly demonstrated rather than just making very general citations...i.e. it would be important to show a PNC that really is stable before transformation by dehydration.

We have changed the organization of our sections to make a more clear discussion of the proposed mechanism. In the original manuscript Section 4 included the discussion on the (a) formation enthalpies, (b) the MD aggregation, and (c) the proposed mechanism. We have moved a and b to section 3 making subsections 3.4, and then keep section 4 only to discuss the mechanism.

Regarding the last suggestion “it would be important to show a PNC that really is stable before transformation by dehydration” we use these months to simulate 2.5 times longer times. But we are working in supersaturated conditions, likely those of the globule itself, so the PPs aggregate fast and we do not prove that PPs are stable. In any case, we think that the dehydration will take place in the aggregate stage, and not in the individual PPs.

References

- [1] A. Carino et al , Cryst. Growth Des., 17 (4), pp 2006–2015, (2017), DOI: 10.1021/acs.cgd.7b00006
- [2] De Yoreo, J.J., et al., Science, 2015. 349(6247).
- [3] J. J. Thomas, et al, Cem. Concr. Res., 2011, 41, 1257–1278.
- [4] G. W. Scherer and F. Bellmann, Cem. Concr. Res., 2017, DOI:10.1016/j.cemconres.2016.07.017.
- [5] G. W. Scherer and F. Bellmann, Cem. Concr. Res., 2017, DOI: 10.1016/j.cemconres.2016.07.017.
- [6] M.R.Andalibi, et al, J.Mater.Chem.A,6, 363-373, 2018. [https://DOI: 10.1039/c7ta08784e](https://doi.org/10.1039/c7ta08784e)
- [7] A. Carino et al, Acta Biomaterialia 74, 478-488, 2018
- [8] Henzler et al., Sci. Adv. 2018;4 (1) – eaao6283 : DOI: 10.1126/sciadv.aao6283
- [9] A. E. Nielsen, Kinetics of Precipitation, Pergamon Press Oxford, 1964.
- [10] A. Testino, et al Chem. Mater., 2005, 17, 5346–5356.
- [11] A. Kumar, et al J. Phys. Chem. C, 2017, 121, 17188–17196.
- [12] S. Galmarini et al, J. Phys. Chem. C 120, 22407-22413 (2016)
- [13] I. Richardson – Acet Cryst.(2014) B70 903-923

- [14] Duque-Redondo, E., et al, Cem. Concr. Res. 156, 106784 (2022).
- [15] Kulik, D. A.; et al. Comput. Geosci.2013, 17, 1–24.
- [16] V. Kanchanason, J. Plank, Cem. Concr. Res. 102 (2017) 90–98.
<https://doi.org/10.1016/J.CEMCONRES.2017.09.002>.

REVIEWERS' COMMENTS

Reviewer #1 (Remarks to the Author):

The authors have sufficiently address my concerns.

Reviewer #3 (Remarks to the Author):

The authors have addressed all the issued raised by the reviewer and made appropriate changes to the manuscript accordingly. They have also done an excellent job framing their work in the ongoing discussion of multistep nucleation and classical versus non-classical nucleation. Overall this is an important work that merits publication in Nature Communication.

Reviewer #4 (Remarks to the Author):

General comments

The authors have taken seriously all reviewers comments and have in my opinion significantly improved the paper consequently. They have clarified all the points raised and modified the manuscript accordingly. I have only spotted a few very minor typos as indicated below. the paper can now be accepted with these minor typos corrected. Congratulations, an excellent scientific study.

Detailed comments

Line 69 dynamics – should read – dynamic

Line 331 – believe that trimmers – should read – believed that trimers

Line 423 Then, a intra- should read – then, an intra-

Line 485 – stabilise – should read – stabilized

Line 493 - 3 the – should read 3 A (angstroms) the

Line 534 uses as- should read – used as a

Line 556- possible – should read - possibly

Line 565 explore – should read – explored

Line 565 predict – should read – predicts